# Beyond the Mean-Field: Structured Deep Gaussian Processes Improve the Predictive Uncertainties

**Jakob Lindinger[1,2,3], David Reeb[1], Christoph Lippert[2,3], Barbara Rakitsch[1]**
[1]Bosch Center for Artificial Intelligence, Renningen, Germany
[2]Hasso Plattner Institute, Potsdam, Germany
[3]University of Potsdam, Germany
{jakob.lindinger, david.reeb, barbara.rakitsch}@de.bosch.com, christoph.lippert@hpi.de

## Abstract

Deep Gaussian Processes learn probabilistic data representations for supervised learning by cascading multiple Gaussian Processes. While this model family promises flexible predictive distributions, exact inference is not tractable. Approximate inference techniques trade off the ability to closely resemble the posterior distribution against speed of convergence and computational efficiency. We propose a novel Gaussian variational family that allows for retaining covariances between latent processes while achieving fast convergence by marginalising out all global latent variables. After providing a proof of how this marginalisation can be done for general covariances, we restrict them to the ones we empirically found to be most important in order to also achieve computational efficiency. We provide an efficient implementation of our new approach and apply it to several benchmark datasets. It yields excellent results and strikes a better balance between accuracy and calibrated uncertainty estimates than its state-of-the-art alternatives.

## 1 Introduction

Gaussian Processes (GPs) provide a non-parametric framework for learning distributions over unknown functions from data [21]: As the posterior distribution can be computed in closed-form, they return well-calibrated uncertainty estimates, making them particularly useful in safety critical applications [3, 22], Bayesian optimisation [10, 30], active learning [37] or under covariate shift [31]. However, the analytical tractability of GPs comes at the price of reduced flexibility: Standard kernel functions make strong assumptions such as stationarity or smoothness. To make GPs more flexible, a practitioner would have to come up with hand-crafted features or kernel functions. Both alternatives require expert knowledge and are prone to overfitting.

Deep Gaussian Processes (DGPs) offer a compelling alternative since they learn non-linear feature representations in a fully probabilistic manner via GP cascades [6]. The gained flexibility has the drawback that inference can no longer be carried out in closed-form, but must be performed via Monte Carlo sampling [9], or approximate inference techniques [5, 6, 24]. The most popular approximation, variational inference, searches for the best approximate posterior within a pre-defined class of distributions: the variational family [4]. For GPs, variational approximations often build on the inducing point framework where a small set of global latent variables acts as pseudo datapoints summarising the training data [29, 32]. For DGPs, each latent GP is governed by its own set of inducing variables, which, in general, need not be independent from those of other latent GPs. Here, we offer a new class of variational families for DGPs taking the following two requirements into account: (i) all global latent variables, i.e., inducing outputs, can be marginalised out, (ii) correlations between latent GP models can be captured. Satisfying (i) reduces the variance in the estimators and is needed for fast convergence [16] while (ii) leads to better calibrated uncertainty estimates [33].

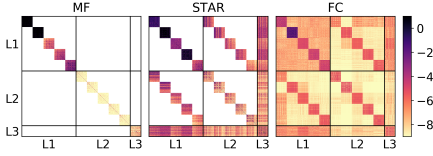

Figure 1: **Covariance matrices for variational posteriors.** We used a DGP with 2 hidden layers (L1, L2) of 5 latent GPs each and a single GP in the output layer (L3). The complexity of the variational approximation is increased by allowing for additional dependencies within and across layers in a Gaussian variational family (left: mean-field [24], middle: stripes-and-arrow, right: fully-coupled). Plotted are natural logarithms of the absolute values of the variational covariance matrices over the inducing outputs.

By using a fully-parameterised Gaussian variational posterior over the global latent variables, we automatically fulfil (ii), and we show in Sec. 3.1, via a proof by induction, that (i) can still be achieved. The proof is constructive, resulting in a novel inference scheme for variational families that allow for correlations within and across layers. The proposed scheme is general and can be used for arbitrarily structured covariances allowing the user to easily adapt it to application-specific covariances, depending on the desired DGP model architecture and on the system requirements with respect to speed, memory and accuracy. One particular case, in which the variational family is chain-structured, has also been considered in a recent work [34], in which the compositional uncertainty in deep GP models is studied.

In Fig. 1 (right) we depict exemplary inferred covariances between the latent GPs for a standard deep GP architecture. In addition to the diagonal blocks, the covariance matrix has visible diagonal stripes in the off-diagonal blocks and an arrow structure. These diagonal stripes point towards strong dependencies between successive latent GPs, while the arrow structure reflects dependencies between all hidden layers and the output layer. In Sec. 3.2, we further propose a scalable approximation to this variational family, which only takes these stronger correlations into account (Fig. 1, middle). We provide efficient implementations for both variational families, where we particularly exploit the sparsity and structure of the covariance matrix of the variational posterior. In Sec. 4, we show experimentally that the new algorithm works well in practice. Our approach obtains a better balance between accurate predictions and calibrated uncertainty estimates than its competitors, as we showcase by varying the distance of the test from the training points.

## 2    Background

In the following, we introduce the notation and provide the necessary background on DGP models. GPs are their building blocks and the starting point of our review.

### 2.1    Primer on Gaussian Processes

In regression problems, the task is to learn a function $f : \mathbb{R}^D \to \mathbb{R}$ that maps a set of $N$ input points $x_N = \{x_n\}_{n=1}^N$ to a corresponding set of noisy outputs $y_N = \{y_n\}_{n=1}^N$. Throughout this work, we assume iid noise, $p(y_N|f_N) = \prod_{n=1}^N p(y_n|f_n)$, where $f_n = f(x_n)$ and $f_N = \{f_n\}_{n=1}^N$ are the function values at the input points. We place a zero mean GP prior on the function $f$, $f \sim \mathcal{GP}(0, k)$, where $k : \mathbb{R}^D \times \mathbb{R}^D \to \mathbb{R}$ is the kernel function. This assumption leads to a multivariate Gaussian prior over the function values, $p(f_N) = \mathcal{N}(f_N|0, K_{NN})$ with covariance matrix $K_{NN} = \{k(x_n, x_{n'})\}_{n,n'=1}^N$.

In preparation for the next section, we introduce a set of $M \ll N$ so-called inducing points $x_M = \{x_m\}_{m=1}^M$ from the input space[1] [29, 32]. From the definition of a GP, the corresponding inducing outputs $f_M = \{f_m\}_{m=1}^M$, where $f_m = f(x_m)$, share a joint multivariate Gaussian distribution with $f_N$. We can therefore write the joint density as $p(f_N, f_M) = p(f_N|f_M)p(f_M)$, where we factorised the joint prior into $p(f_M) = \mathcal{N}(f_M|0, K_{MM})$, the prior over the inducing outputs, and the conditional $p(f_N|f_M) = \mathcal{N}\left(f_N\middle|\widetilde{K}_{NM}f_M, \widetilde{K}_{NN}\right)$ with

$$\widetilde{K}_{NM} = K_{NM}\left(K_{MM}\right)^{-1}, \quad \widetilde{K}_{NN} = K_{NN} - K_{NM}\left(K_{MM}\right)^{-1}K_{MN}. \tag{1}$$

Here the matrices $K$ are defined similarly as $K_{NN}$ above, e.g. $K_{NM} = \{k(x_n, x_m)\}_{n,m=1}^{N,M}$.

## 2.2 Deep Gaussian Processes

A deep Gaussian Process (DGP) is a hierarchical composition of GP models. We consider a model with $L$ layers and $T_l$ (stochastic) functions in layer $l = 1, \ldots, L$, i.e., a total number of $T = \sum_{l=1}^{L} T_l$ functions [6]. The input of layer $l$ is the output of the previous layer, $f_N^l = [f^{l,1}(f_N^{l-1}), \ldots, f^{l,T_l}(f_N^{l-1})]$, with starting values $f_N^0 = x_N$. We place independent GP priors augmented with inducing points on all the functions, using the same kernel $k^l$ and the same set of inducing points $x_M^l$ within layer $l$. This leads to the following joint model density:

$$p(y_N, f_N, f_M) = p(y_N|f_N^L) \prod_{l=1}^{L} p(f_N^l|f_M^l; f_N^{l-1}) p(f_M^l). \tag{2}$$

Here $p(f_M^l) = \prod_{t=1}^{T_l} \mathcal{N}\left(f_M^{l,t}\middle|0, K_{MM}^l\right)$ and $p(f_N^l|f_M^l; f_N^{l-1}) = \prod_{t=1}^{T_l} \mathcal{N}\left(f_N^{l,t}\middle|\widetilde{K}_{NM}^l f_M^{l,t}, \widetilde{K}_{NN}^l\right)$, where $\widetilde{K}_{NM}^l$ and $\widetilde{K}_{NN}^l$ are given by the equivalents of Eq. (1), respectively.[2]

Inference in this model (2) is intractable since we cannot marginalise over the latents $f_N^1, \ldots, f_N^{L-1}$ as they act as inputs to the non-linear kernel function. We therefore choose to approximate the posterior by employing variational inference: We search for an approximation $q(f_N, f_M)$ to the true posterior $p(f_N, f_M|y_N)$ by first choosing a variational family for the distribution $q$ and then finding an optimal $q$ within that family that minimises the Kullback-Leibler (KL) divergence $\mathrm{KL}[q||p]$. Equivalently, the so-called evidence lower bound (ELBO),

$$\mathcal{L} = \int q(f_N, f_M) \log \frac{p(y_N, f_N, f_M)}{q(f_N, f_M)} df_N df_M, \tag{3}$$

can be maximised. In the following, we choose the variational family [24]

$$q(f_N, f_M) = q(f_M) \prod_{l=1}^{L} p(f_N^l|f_M^l; f_N^{l-1}). \tag{4}$$

Note that $f_M = \{f_M^{l,t}\}_{l,t=1}^{L,T_l}$ contains the inducing outputs of all layers, which might be covarying. This observation will be the starting point for our structured approximation in Sec. 3.1.

In the remaining part of this section, we follow Ref. [24] and restrict the distribution over the inducing outputs to be a-posteriori Gaussian and independent between different GPs (known as mean-field assumption, see also Fig. 1, left), $q(f_M) = \prod_{l=1}^{L} \prod_{t=1}^{T_l} q(f_M^{l,t})$. Here $q(f_M^{l,t}) = \mathcal{N}\left(f_M^{l,t}\middle|\mu_M^{l,t}, S_M^{l,t}\right)$ and $\mu_M^{l,t}, S_M^{l,t}$ are free variational parameters. The inducing outputs $f_M$ act thereby as global latent variables that capture the information of the training data. Plugging $q(f_M)$ into Eqs. (2), (3), (4), we can simplify the ELBO to

$$\mathcal{L} = \sum_{n=1}^{N} \mathbb{E}_{q(f_n^L)}\left[\log p(y_n|f_n^L)\right] - \sum_{l,t=1}^{L,T_l} \mathrm{KL}[q(f_M^{l,t})||p(f_M^{l,t})]. \tag{5}$$

We first note that the ELBO decomposes over the data points, allowing for minibatch subsampling [12]. However, the marginals of the output of the final layer, $q(f_n^L)$, cannot be obtained analytically. While the mean-field assumption renders it easy to analytically marginalise out the inducing outputs (see Appx. D.1), the outputs of the intermediate layers cannot be fully integrated out, since they are kernel inputs of the respective next layer, leaving us with

$$q(f_n^L) = \int \prod_{l=1}^{L} q(f_n^l; f_n^{l-1}) df_n^1 \cdots df_n^{L-1}, \text{ where } q(f_n^l; f_n^{l-1}) = \prod_{t=1}^{T_l} \mathcal{N}\left(f_n^{l,t}\middle|\widetilde{\mu}_n^{l,t}, \widetilde{\Sigma}_n^{l,t}\right). \tag{6}$$

The means and covariances are given by

$$\widetilde{\mu}_n^{l,t} = \widetilde{K}_{nM}^l \mu_M^{l,t}, \quad \widetilde{\Sigma}_n^{l,t} = K_{nn}^l - \widetilde{K}_{nM}^l\left(K_{MM}^l - S_M^{l,t}\right)\widetilde{K}_{Mn}^l. \tag{7}$$

We can straightforwardly obtain samples from $q(f_n^L)$ by recursively sampling through the layers using Eq. (6). Those samples can be used to evaluate the ELBO [Eq. (5)] and to obtain unbiased gradients for parameter optimisation by using the reparameterisation trick [16, 23]. This stochastic estimator of the ELBO has low variance as we only need to sample over the local latent parameters $f_n^1, \ldots, f_n^{L-1}$, while we can marginalise out the global latent parameters, i.e. inducing outputs, $f_M$.

## 3    Structured Deep Gaussian Processes

Next, we introduce a new class of variational families that allows to couple the inducing outputs $f_M$ within and across layers. Surprisingly, analytical marginalisation over the inducing outputs $f_M$ is still possible after reformulating the problem into a recursive one that can be solved by induction. This enables an efficient inference scheme that refrains from sampling any global latent variables. Our method generalises to arbitrary interactions which we exploit in the second part where we focus on the most prominent ones to attain speed-ups.

### 3.1    Fully-Coupled DGPs

We present now a new variational family that offers both, efficient computations and expressivity: Our approach is efficient, since all global latent variables can be marginalised out , and expressive, since we allow for structure in the variational posterior. We do this by leaving the Gaussianity assumption unchanged, while permitting dependencies between all inducing outputs (within layers and also across layers). This corresponds to the (variational) ansatz $q(f_M) = \mathcal{N}(f_M|\mu_M, S_M)$ with dimensionality $TM$. By taking the dependencies between the latent processes into account, the resulting variational posterior $q(f_N, f_M)$ [Eq. (4)] is better suited to closely approximate the true posterior. We give a comparison of exemplary covariance matrices $S_M$ in Fig. 1.

Next, we investigate how the ELBO computations have to be adjusted when using the fully-coupled variational family. Plugging $q(f_M)$ into Eqs. (2), (3) and (4), yields

$$\mathcal{L} = \sum_{n=1}^{N} \mathbb{E}_{q(f_n^L)} \left[ \log p(y_n|f_n^L) \right] - \text{KL}[q(f_M)|| \prod_{l,t=1}^{L,T_l} p(f_M^{l,t})], \tag{8}$$

which we derive in detail in Appx. C. The major difference to the mean-field DGP lies in the marginals $q(f_n^L)$ of the outputs of the last layer: Assuming (as in the mean-field DGP) that the distribution over the inducing outputs $f_M$ factorises between the different GPs causes the marginalisation integral to factorise into $L$ standard Gaussian integrals. This is not the case for the fully-coupled DGP (see Appx. D.1 for more details), which makes the computations more challenging. The implications of using a fully coupled $q(f_M)$ are summarised in the following theorem.

**Theorem 1.** *In a fully-coupled DGP as defined above, the marginals $q(f_n^L)$ can be written as*

$$q(f_n^L) = \int \prod_{l=1}^{L} q(f_n^l|f_n^1, \ldots, f_n^{l-1}) df_n^1 \cdots df_n^{L-1} \text{ where } q(f_n^l|f_n^1, \ldots, f_n^{l-1}) = \mathcal{N}\left( f_n^l \Big| \hat{\mu}_n^l, \hat{\Sigma}_n^l \right), \tag{9}$$

*for each data point $x_n$. The means and covariances are given by*

$$\hat{\mu}_n^l = \widetilde{\mu}_n^l + \widetilde{S}_n^{l,1:l-1} \left( \widetilde{S}_n^{1:l-1,1:l-1} \right)^{-1} (f_n^{1:l-1} - \widetilde{\mu}_n^{1:l-1}), \tag{10}$$

$$\hat{\Sigma}_n^l = \widetilde{S}_n^{ll} - \widetilde{S}_n^{l,1:l-1} \left( \widetilde{S}_n^{1:l-1,1:l-1} \right)^{-1} \widetilde{S}_n^{1:l-1,l}, \tag{11}$$

*where $\widetilde{\mu}_n^l = \widetilde{\mathcal{K}}_{nM}^l \mu_M^l$ and $\widetilde{S}_n^{ll'} = \delta_{ll'}\mathcal{K}_{nn}^l - \widetilde{\mathcal{K}}_{nM}^l \left( \delta_{ll'}\mathcal{K}_{MM}^l - S_M^{ll'} \right) \widetilde{\mathcal{K}}_{Mn}^{l'}$.*

In Eqs. (10) and (11) the notation $A^{l,1:l'}$ is used to index a submatrix of the variable $A$, e.g. $A^{l,1:l'} = \left( A^{l,1} \cdots A^{l,l'} \right)$. Additionally, $\mu_M^l \in \mathbb{R}^{T_l M}$ denotes the subvector of $\mu_M$ that contains the means of the inducing outputs in layer $l$, and $S_M^{ll'} \in \mathbb{R}^{T_l M \times T_{l'} M}$ contains the covariances between the inducing outputs of layers $l$ and $l'$. For $\widetilde{\mu}_n^l$ and $\widetilde{S}_n^{ll'}$, we introduced the notation $\mathcal{K}^l = \left( \mathbb{I}_{T_l} \otimes K^l \right)$ as

shorthand for the Kronecker product between the identity matrix $\mathbb{I}_{T_l}$ and the covariance matrix $K^l$, and used $\delta$ for the Kronecker delta. We verify in Appx. B.2 that the formulas contain the mean-field solution as a special case by plugging in the respective covariance matrix.

By Thm. 1, the inducing outputs $f_M$ can still be marginalised out, which enables low-variance estimators of the ELBO. While the resulting formula for $q(f_n^l|f_n^1,\ldots,f_n^{l-1})$ has a similar form as Gaussian conditionals, this is only true at first glance (cf. also Appx. B.1): The latents of the preceding layers $f_n^{1:l-1}$ enter the mean $\hat{\mu}_n^l$ and the covariance matrix $\hat{\Sigma}_n^l$ also in an indirect way via $\widetilde{S}_n$ as they appear as inputs to the kernel matrices.

**Sketch of the proof of Theorem 1**. We start the proof with the general formula for $q(f_n^L)$,

$$q(f_n^L) = \int \left[ \int q(f_M) \prod_{l'=1}^{L} p(f_n^{l'}|f_M^{l'}) df_M \right] df_n^1 \cdots df_n^{L-1}, \tag{12}$$

which is already (implicitly) used in Ref. [24] and which we derive in Appx. D. In order to show the equivalence between the inner integral in Eq. (12) and the integrand in Eq. (9) we proceed to find a recursive formula for integrating out the inducing outputs layer after layer:

$$\int q(f_M) \prod_{l'=1}^{L} p(f_n^{l'}|f_M^{l'}) df_M = \left[ \prod_{l'=1}^{l-1} q(f_n^{l'}|f_n^{1:l'-1}) \right] \int q(f_n^l, f_M^{l+1:L}|f_n^{1:l-1}) \prod_{l'=l+1}^{L} p(f_n^{l'}|f_M^{l'}) df_M^{l'}. \tag{13}$$

The equation above holds for $l = 1,\ldots,L$ after the inducing outputs of layers $1,\ldots,l$ have already been marginalised out. This is stated more formally in Lem. 2 in Appx. A, in which we also provide exact formulas for all terms. Importantly, all of them are multivariate Gaussians with known mean and covariance. The lemma itself can be proved by induction and we will show the general idea of the induction step here: For this, we assume the right hand side of Eq. (13) to hold for some layer $l$ and then prove that it also holds for $l \to l+1$. We start by taking the (known) distribution within the integral and split it in two by conditioning on $f_n^l$:

$$q(f_n^l, f_M^{l+1:L}|f_n^{1:l-1}) = q(f_n^l|f_n^{1:l-1})q(f_M^{l+1:L}|f_n^{1:l}) \tag{14}$$

Then we show that the distribution $q(f_n^l|f_n^{1:l-1})$ can be written as part of the product in front of the integral in Eq. (13) (thereby increasing the upper limit of the product to $l$). Next, we consider the integration over $f_M^{l+1}$, where we collect all relevant terms (thereby increasing the lower limit of the product within the integral in Eq. (13) to $l+2$):

$$\int q(f_M^{l+1:L}|f_n^{1:l})p(f_n^{l+1}|f_M^{l+1})df_M^{l+1} = \int q(f_M^{l+1}|f_n^{1:l})q(f_M^{l+2:L}|f_n^{1:l},f_M^{l+1})p(f_n^{l+1}|f_M^{l+1})df_M^{l+1}$$

$$= \int q(f_M^{l+1}|f_n^{1:l})q(f_n^{l+1},f_M^{l+2:L}|f_n^{1:l},f_M^{l+1})df_M^{l+1} = q(f_n^{l+1},f_M^{l+2:L}|f_n^{1:l}). \tag{15}$$

The terms in the first line are given by Eqs. (14) and (2). All subsequent terms are also multivariate Gaussians that are obtained by standard operations like conditioning, joining two distributions, and marginalisation. We can therefore give an analytical expression of the final term in Eq. (15), which is exactly the term that is needed on the right hand side of Eq. (13) for $l \to l+1$. Confirming that this term has the correct mean and covariance completes the induction step.

After proving Lem. 2, Eq. (13) can be used. For the case $l = L$ the right hand side can be shown to yield $\prod_{l=1}^{L} q(f_n^l|f_n^1,\ldots,f_n^{l-1})$. Hence, Eq. (9) follows by substituting the inner integral in Eq. (12) by this term. The full proof can be found in Appx. A. □

Furthermore, we give a heuristic argument for Thm. 1 in Appx. B.1 in which we show that by ignoring the recursive structure of the prior, the marginalisation of the inducing outputs $f_M$ becomes straightforward. While mathematically not rigorous, the derivation provides additional intuition.

Next, we use our novel variational approach to fit a fully coupled DGP model with $L = 3$ layers to the *concrete* UCI dataset. We can clearly observe that this algorithmic work pays off: Fig. 1 shows that there is more structure in the covariance matrix $S_M$ than the mean-field approximation allows. This additional structure results in a better approximation of the true posterior as we validate on a

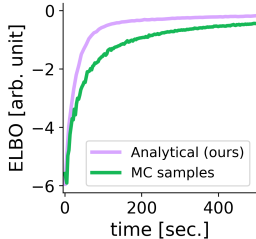

Figure 2: **Convergence behaviour: Analytical vs. MC marginalisation.** We plot the ELBO as a function of time in seconds when the marginalisation of the inducing outputs $f_M$ is performed analytically via our Thm. 1 (purple) and via MC sampling (green). We used a fully-coupled DGP with our standard three layer architecture (see Sec. 3.2), on the *concrete* UCI dataset trained with Adam [15].

range of benchmark datasets (see Tab. S5 in Appx. G) for which we observe larger ELBO values for the fully-coupled DGP than for the mean-field DGP. Additionally, we show in Fig. 2 that our analytical marginalisation over the inducing outputs $f_M$ leads to faster convergence compared to Monte Carlo (MC) sampling, since the corresponding ELBO estimates have lower variance. Independently from our work, the sampling-based approach has also been proposed in Ref. [34].

However, in comparison with the mean-field DGP, the increase in the number of variational parameters also leads to an increase in runtime and made convergence with standard optimisers fragile due to many local optima. We were able to circumvent the latter by the use of natural gradients [2], which have been found to work well for (D)GP models before [10, 26, 25], but this increases the runtime even further (see Sec. 4.2). It is therefore necessary to find a smaller variational family if we want to use the method in large-scale applications.

An optimal variational family combines the best of both worlds, i.e., being as efficient as the mean-field DGP while retaining the most important interactions introduced in the fully-coupled DGP. We want to emphasise that there are many possible ways of restricting the covariance matrix $S_M$ that potentially lead to benefits in different applications. For example, the recent work [34] studies the compositional uncertainty in deep GPs using a particular restriction of the inverse covariance matrix. The authors also provide specialised algorithms to marginalise out the inducing outputs in their model. Here, we provide an analytic marginalisation scheme for arbitrarily structured covariance matrices that will vastly simplify future development of application-specific covariances. Through the general framework that we have developed, testing them is straightforward and can be done via simply implementing a naive version of the covariance matrix in our code.[3] In the following, we propose one possible class of covariance matrices based on our empirical findings.

### 3.2 Stripes-and-Arrow Approximation

In this section, we describe a new variational family that trades off efficiency and expressivity by sparsifying the covariance matrix $S_M$. Inspecting Fig. 1 (right) again, we observe that besides the $M \times M$ blocks on the diagonal, the diagonal stripes [28] (covariances between the GPs in latent layers at the same relative position), and an arrow structure (covariances from every intermediate layer GP to the output GP) receive large values. We make similar observations also for different datasets and different DGP architectures as shown in Fig. S8 in Appx. G. Note that the stripes pattern can also be motivated theoretically as we expect the residual connections realised by the mean functions (footnote 2) to lead to a coupling between successive latent GPs. We therefore propose as one special form to keep only these terms and neglect all other dependencies by setting them to zero in the covariance matrix, resulting in a structure consisting of an arrowhead and diagonal stripes (see Fig. 1 middle).

Denoting the number of GPs per latent layer as $\tau$, it is straightforward to show that the number of non-zero elements in the covariance matrices of mean-field DGP, stripes-and-arrow DGP, and fully-coupled DGP scale as $\mathcal{O}(\tau L M^2)$, $\mathcal{O}(\tau L^2 M^2)$, and $\mathcal{O}(\tau^2 L^2 M^2)$, respectively. In the example of Fig. 1, we have used $\tau = 5$, $L = 3$, and $M = 128$, yielding $1.8 \times 10^5$, $5.1 \times 10^5$, and $2.0 \times 10^6$ non-zero elements in the covariance matrices. Reducing the number of parameters already leads to shorter training times since less gradients need to be computed. Furthermore, the property that makes this form so compelling is that the covariance matrix $\widetilde{S}_n^{1:l-1,1:l-1}$ [needed in Eqs. (10) and

(11)] as well as the Cholesky decomposition[4] of $S_M$ have the same sparsity pattern. Therefore only the non-zero elements at pre-defined positions have to be calculated which is explained in Appx. E. The complexity for the ELBO is $\mathcal{O}(NM^2\tau L^2 + N\tau^3 L^3 + M^3\tau L^3)$. This is a moderate increase compared to the mean-field DGP whose ELBO has complexity $\mathcal{O}(NM^2\tau L)$, while it is a clear improvement over the fully-coupled approach with complexity $\mathcal{O}(NM^2\tau^2 L^2 + N\tau^3 L^3 + M^3\tau^3 L^3)$ (see Appx. E for derivations). An empirical runtime comparison is provided in Sec. 4.2.

After having discussed the advantages of the proposed approximation a remark on a disadvantage is in order: The efficient implementation of Ref. [26] for natural gradients cannot be used in this setting, since the transformation from our parameterisation to a fully-parameterised multivariate Gaussian is not invertible. However, this is only a slight disadvantage since the stripes-and-arrow approximation has a drastically reduced number of parameters, compared to the fully-coupled approach, and we experimentally do not observe the same convergence problems when using standard optimisers (see Appx. G, Fig. S5).

### 3.3 Joint sampling of global and local latent variables

In contrast to our work, Refs. [9, 36] drop the Gaussian assumption over the inducing outputs $f_M$ and allow instead for potentially multi-modal approximate posteriors. While their approaches are arguably more expressive than ours, their flexibility comes at a price: the distribution over the inducing outputs $f_M$ is only given implicitly in form of Monte Carlo samples. Since the inducing outputs $f_M$ act as global latent parameters, the noise attached to their sampling-based estimates affects all samples from one mini-batch. This can often lead to higher variances which may translate to slower convergence [16]. We compare to Ref. [9] in our experiments.

## 4 Experiments

In Sec. 4.1, we study the predictive performance of our stripes-and-arrow approximation. Since it is difficult to assess accuracy and calibration on the same task, we ran a joint study of interpolation and extrapolation tasks, where in the latter the test points are distant from the training points. We found that the proposed approach balances accuracy and calibration, thereby outperforming its competitors on the combined task. Examining the results for the extrapolation task more closely, we find that our proposed method significantly outperforms the competing DGP approaches. In Sec. 4.2, we assess the runtime of our methods and confirm that our approximation has only a negligible overhead compared to mean-field and is more efficient than a fully-coupled DGP. Due to space constraints, we moved many of the experimental details to Appx. G.

### 4.1 Benchmark Results

We compared the predictive performance of our efficient stripes-and-arrow approximation (STAR DGP) with a mean-field approximation (MF DGP) [24], stochastic gradient Hamiltonian Monte Carlo (SGHMC DGP) [9] and a sparse GP (SGP) [12]. As done in prior work, we report results on eight UCI datasets and employ as evaluation criterion the average marginal test log-likelihood (tll).

We assessed the interpolation behaviour of the different approaches by randomly partitioning the data into a training and a test set with a $90 : 10$ split. To investigate the extrapolation behaviour, we created test instances that are distant from the training samples: We first randomly projected the inputs $X$ onto a one-dimensional subspace $z = Xw$, where the weights $w \in \mathbb{R}^D$ were drawn from a standard Gaussian distribution. We subsequently ordered the samples w.r.t. $z$ and divided them accordingly into training and test set using a $50 : 50$ split.

We first confirmed the reports from the literature [9, 24], that DGPs have on interpolation tasks an improved performance compared to sparse GPs (Tab. 1). We also observed that in this setting SGHMC outperforms the MF DGP and our method, which are on par.

Subsequently, we performed the same analysis on the extrapolation task. While our approach, STAR DGP, seems to perform slightly better than MF DGP and also SGHMC DGP, the large standard errors of all methods hamper a direct comparison (see Tab. S3 in Appx. G). This is mainly due to the

Table 1: **Interpolation behaviour on UCI benchmark datasets.** We report marginal tlls (the larger, the better) for various methods, where $L$ denotes the number of layers. Standard errors are obtained by repeating the experiment 10 times. We marked all methods in bold that performed better or as good as the standard sparse GP.

| Dataset (N,D) | SGP L1 | SGHMC DGP L1 | L2 | L3 | MF DGP L2 | L3 | STAR DGP L2 | L3 |
|---|---|---|---|---|---|---|---|---|
| boston (506,13) | -2.58(0.10) | -2.75(0.18) | **-2.51(0.07)** | **-2.53(0.09)** | **-2.43(0.05)** | **-2.48(0.06)** | **-2.47(0.08)** | **-2.43(0.05)** |
| energy (768, 8) | -0.71(0.03) | -1.16(0.44) | **-0.37(0.12)** | **-0.34(0.11)** | **-0.73(0.02)** | -0.75(0.02) | -0.75(0.02) | -0.75(0.02) |
| concrete (1030, 8) | -3.09(0.02) | -3.50(0.34) | **-2.89(0.06)** | **-2.88(0.06)** | **-3.06(0.03)** | **-3.09(0.02)** | **-3.04(0.02)** | **-3.05(0.02)** |
| wine red (1599,11) | -0.88(0.01) | -0.90(0.03) | **-0.81(0.03)** | **-0.80(0.07)** | -0.89(0.01) | -0.89(0.01) | **-0.88(0.01)** | **-0.88(0.01)** |
| kin8nm (8192, 8) | 1.05(0.01) | **1.14(0.01)** | **1.38(0.01)** | **1.25(0.14)** | **1.30(0.01)** | **1.31(0.01)** | **1.28(0.01)** | **1.29(0.01)** |
| power (9568, 4) | -2.78(0.01) | **-2.75(0.02)** | **-2.68(0.02)** | **-2.65(0.02)** | **-2.77(0.01)** | **-2.76(0.01)** | **-2.77(0.01)** | **-2.77(0.01)** |
| naval (11934,16) | 7.56(0.09) | **7.77(0.04)** | 7.32(0.02) | 6.89(0.43) | 7.11(0.11) | 7.05(0.09) | 7.06(0.08) | 6.25(0.31) |
| protein (45730, 9) | -2.91(0.00) | **-2.76(0.00)** | **-2.64(0.01)** | **-2.58(0.01)** | **-2.83(0.00)** | **-2.79(0.00)** | **-2.83(0.00)** | **-2.80(0.00)** |

| Dataset | MF vs. STAR | SGHMC vs. STAR |
|---|---|---|
| boston | **0.55(0.04)** | 0.50(0.05) |
| energy | **0.73(0.05)** | **0.60(0.04)** |
| concrete | **0.57(0.04)** | **0.60(0.03)** |
| wine red | **0.57(0.04)** | **0.63(0.02)** |
| kin8nm | *0.36(0.03)* | *0.44(0.05)* |
| power | 0.44(0.06) | **0.64(0.03)** |
| naval | **0.67(0.06)** | **0.58(0.03)** |
| protein | 0.49(0.03) | 0.50(0.03) |

Table 2: **Extrapolation behaviour: direct comparison of DGP methods.** Average frequency $\mu$ and its standard error $\sigma$ (computed over 10 repetitions) of the STAR DGP outperforming the MF DGP (left) and the SGHMC DGP (right) on the marginal tll of individual repetitions of the extrapolation task (see main text for details). Results are for DGPs with three layers. We mark numbers in bold (italics) if STAR outperforms its competitor (vice versa).

random 1D-projection of the extrapolation experiment: The direction of the projection has a large impact on the difficulty of the prediction task. Since this direction changes over the repetitions, the corresponding test log-likelihoods vary considerably, leading to large standard errors.

We resolved this issue by performing a direct comparison between STAR DGP and the other two DGP variants: To do so, we computed the frequency of test samples for which STAR DGP obtained a larger log-likelihood than MF/SGHMC DGP on each train-test split independently. Average frequency $\mu$ and its standard error $\sigma$ were subsequently computed over 10 repetitions and are reported in Tab. 2. On 5/8 datasets STAR DGP significantly outperforms MF DGP and SGHMC DGP ($\mu > 0.50 + \sigma$), respectively, while the opposite only occurred on *kin8nm*. In Tab. S4 in Appx. G, we show more comparisons, that also take the absolute differences in test log likelihoods into account and additionally consider the comparison of fully-coupled and MF DGP. Taken together, we conclude that our structured approximations are in particular beneficial in the extrapolation scenario, while their performance is similar to MF DGP in the interpolation scenario.

Next, we performed an in-depth comparison between the approaches that analytically marginalise the inducing outputs: In Fig. 3 we show that the predicted variance $\sigma_*^2$ increased as we moved away from the training data (left) while the mean squared errors also grew with larger $\sigma_*^2$ (right). The mean squared error is an empirical unbiased estimator of the variance $\text{Var}_* = \mathbb{E}[(y_* - \mu_*)^2]$ where $y_*$ is the test output and $\mu_*$ the mean predictor. The predicted variance $\sigma_*^2$ is also an estimator of $\text{Var}_*$. It is only unbiased if the method is calibrated. However, we observed for the mean-field approach that, when moving away from the training data, the mean squared error was larger than the predicted variances pointing towards underestimated uncertainties. While the mean squared error for SGP matched well with the predictive variances, the predictions are rather inaccurate as demonstrated by the large predicted variances. Our method reaches a good balance, having generally more accurate mean predictions than SGP and at the same time more accurate variance predictions than MF DGP.

Finally, we investigated the behaviour of the SGHMC approaches in more detail. We first ran a one-layer model that is equivalent to a sparse GP but with a different inference scheme: Instead of marginalising out the inducing outputs, they are sampled. We observed that the distribution over the inducing outputs is non-Gaussian (see Appx. G, Fig. S6), even though the optimal approximate posterior distribution is provably Gaussian in this case [32]. A possible explanation for this are convergence problems since the global latent variables are not marginalised out, which, in turn, offers a potential explanation for the poor extrapolation behaviour of SGHMC that we observed in our experiments across different architectures and datasets. Similar convergence problems have also been observed by Ref. [25].

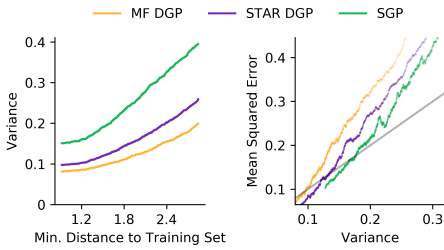

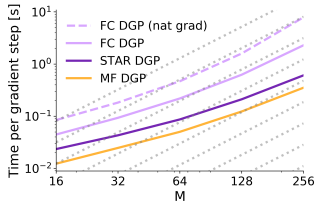

Figure 3: **Calibration Study.** Left: While the predicted variances increase for all methods as a function of the distance to the training data, we find that at any given distance, the uncertainty decreases from SGP to STAR DGP to MF DGP. Right: We plot the mean squared error as a function of the predicted variance. If the mean squared error is larger than the predicted variance, the latter underestimates the uncertainty. Results are recorded on the *kin8nm* UCI dataset and smoothed for plotting by using a median filter.

Figure 4: **Runtime comparison.** We compare the runtime of our efficient STAR DGP versus the FC DGP and the MF DGP on the *protein* UCI dataset. Shown is the runtime of one gradient step in seconds on a logarithmic scale as a function of the number of inducing points $M$. The dotted grey lines show the theoretical runtime $\mathcal{O}(M^2)$.

## 4.2 Runtime

We compared the impact of the variational family on the runtime as a function of the number of inducing points $M$. For the fully-coupled (FC) variational model, we also recorded the runtime when employing natural gradients [26]. The results can be seen in Fig. 4, where the order from fastest to slowest method was proportional to the complexity of the variational family: mean-field, stripes-and-arrow, fully-coupled DGP. For our standard setting, $M = 128$, our STAR approximation was only two times slower than the mean-field but three times faster than FC DGP (trained with Adam [15]). This ratio stayed almost constant when the number of inducing outputs $M$ was changed, since the most important term in the computational costs scales as $\mathcal{O}(M^2)$ for all methods. Subsequently, we performed additional experiments in which we varied the architecture parameters $L$ and $\tau$. Both confirm that the empirical runtime performance scales with the complexity of the variational family (see Appx. G, Fig. S7) and matches our theoretical estimates in Sec. 3.2.

## 5 Summary

In this paper, we investigated a new class of variational families for deep Gaussian processes (GPs). Our approach is (i) efficient as it allows to marginalise analytically over the global latent variables and (ii) expressive as it couples the inducing outputs across layers in the variational posterior. Naively coupling all inducing outputs does not scale to large datasets, hence we suggest a sparse and structured approximation that only takes the most important dependencies into account. In a joint study of interpolation and extrapolation tasks as well as in a careful evaluation of the extrapolation task on its own, our approach outperforms its competitors, since it balances accurate predictions and calibrated uncertainty estimates. Further research is required to understand why our structured approximations are especially helpful for the extrapolation task. One promising direction could be to look at differences of inner layer outputs (as done in Ref. [34]) and link them to the final deep GP outputs.

There has been a lot of follow-up work on deep GPs in which the probabilistic model is altered to allow for multiple outputs [14], multiple input sources [8], latent features [25] or for interpreting the latent states as differential flows [11]. Our approach can be easily adapted to any of these models and is therefore a promising line of work to advance inference in deep GP models.

Our proposed structural approximation is only one way of coupling the latent GPs. Discovering new variational families that allow for more speed-ups either by applying Kronecker factorisations as done in the context of neural networks [18], placing a grid structure over the inducing inputs [13], or by taking a conjugate gradient perspective on the objective [35] are interesting directions for future research. Furthermore, we think that the dependence of the optimal structural approximation on various factors (model architecture, data properties, etc.) is worthwhile to be studied in more detail.

## Broader Impact

In many applications, machine learning algorithms have been shown to achieve superior predictive performance compared to hand-crafted or expert solutions [27]. However, these methods can be applied in safety-critical applications only if they return predictive distributions allowing to quantify the uncertainty of the prediction [17]. For instance, a medical diagnosis tool can be applied only if each diagnosis is endowed with a confidence interval such that in case of ambiguity a physician can be contacted. Our work yields accurate predictive distributions for deep non-parametric models by allowing correlations between and across layers in the variational posterior. As we validate in our experiments, this also holds true when the input distribution at test time differs from the input distribution at training time. In our medical example, this might be the case if the hospital where the data is recorded is different from the one where the diagnosis tool is deployed.

## Acknowledgements and Disclosure of Funding

We thank Buote Xu for valuable comments and suggestions on an early draft of the paper. We furthermore acknowledge the detailed and constructive feedback from the four anonymous reviewers, particularly for suggesting a new experiment which lead to Fig. 2. We have no funding to disclose for this work.

## Footnotes

[1]Note that in our notation variables with an index $m$, $M$ ($n$, $N$) denote quantities related to inducing (training) points. This implies for example that $x_{m=1}$ and $x_{n=1}$ are in general not the same.

[2]In order to avoid pathologies created by highly non-injective mappings in the DGP [7], we follow Ref. [24] and add non-trainable linear mean terms given by the PCA mapping of the input data to the latent layers. Those terms are omitted from the notation for better readability.

[3]Python code (building on code for the mean-field DGP [25], GPflow [19] and TensorFlow [1]) implementing our method is provided at `https://github.com/boschresearch/Structured_DGP`. A pseudocode description of our algorithm is given in Appx. F.

[4]In order to ensure that $S_M$ is positive definite, we will numerically exclusively work with its Cholesky factor $L$, a unique lower triangular matrix such that $S_M = LL^\top$.

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
