[Supplementary Material]

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

## A  Marginalisation of the inducing outputs (proof of Theorem 1)

The aim of this section is to provide a complete proof for Thm. 1. We will do this by starting from the formula for $q(f_n^l)$ that we work out in Appx. D,

$$q(f_n^L) = \int \left[ \int q(f_M) \prod_{l=1}^{L} p(f_n^l | f_M^l; f_n^{l-1}) df_M \right] df_n^1 \cdots df_n^{L-1}. \tag{16}$$

Comparing to Eq. (9), we see that it remains to be shown that indeed

$$\int q(f_M) \prod_{l=1}^{L} p(f_n^l | f_M^l; f_n^{l-1}) df_M = \prod_{l=1}^{L} q(f_n^l | f_n^1, \ldots, f_n^{l-1}), \tag{17}$$

where the distributions $q$ on the right hand side have the properties described in Eqs. (9) - (11). The terms appearing on the left hand side are given by $q(f_M) = \mathcal{N}(f_M | \mu_M, S_M)$, which is interchangeably also denoted as

$$q(f_M^{1:L}) = \mathcal{N}\left( f_M^{1:L} \middle| \mu_M^{1:L}, S_M^{1:L,1:L} \right) = q\left( \begin{pmatrix} f_M^1 \\ \vdots \\ f_M^L \end{pmatrix} \right) = \mathcal{N}\left( \begin{pmatrix} f_M^1 \\ \vdots \\ f_M^L \end{pmatrix} \middle| \begin{pmatrix} \mu_M^1 \\ \vdots \\ \mu_M^L \end{pmatrix}, \begin{pmatrix} S_M^{11} & \cdots & S_M^{1L} \\ \vdots & \ddots & \vdots \\ S_M^{L1} & \cdots & S_M^{LL} \end{pmatrix} \right), \tag{18}$$

and

$$p(f_n^l | f_M^l; f_n^{l-1}) = \mathcal{N}\left( f_n^l \middle| \widetilde{\mathcal{K}}_{nM}^l f_M^l, \widetilde{\mathcal{K}}_{nn}^l \right), \tag{19}$$

where

$$\widetilde{\mathcal{K}}_{nM}^l = \mathcal{K}_{nM}^l \left( \mathcal{K}_{MM}^l \right)^{-1} \tag{20}$$

$$\widetilde{\mathcal{K}}_{nn}^l = \mathcal{K}_{nn}^l - \mathcal{K}_{nM}^l \left( \mathcal{K}_{MM}^l \right)^{-1} \mathcal{K}_{Mn}^l. \tag{21}$$

In order to show that Eq. (17) holds, we will introduce a rather technical lemma in the following and prove it later by induction.

**Lemma 2.** *Given the definitions in Eqs. (18) and (19), $\forall l = 1, \ldots, L$ we have*

$$\int q(f_M^{1:L}) \prod_{l'=1}^{L} p(f_n^{l'} | f_M^{l'}) df_M^{l'} = \left[ \prod_{l'=1}^{l-1} q(f_n^{l'} | f_n^{1:l'-1}) \right] \int q(f_n^l, f_M^{l+1:L} | f_n^{1:l-1}) \prod_{l'=l+1}^{L} p(f_n^{l'} | f_M^{l'}) df_M^{l'}, \tag{22}$$

*where $q(f_n^{l'} | f_n^{1:l'-1})$ is as in Eq. (9) and*

$$q(f_n^l, f_M^{l+1:L} | f_n^{1:l-1}) = \mathcal{N}\left( \begin{pmatrix} f_n^l \\ f_M^{l+1:L} \end{pmatrix} \middle| \begin{pmatrix} \hat{\mu}_n^l \\ {}^l\hat{\mu}_M^{l+1:L} \end{pmatrix}, \begin{pmatrix} \hat{\Sigma}_n^l & {}^l\hat{\Sigma}_{nM}^{l,l+1:L} \\ \left( {}^l\hat{\Sigma}_{nM}^{l,l+1:L} \right)^\top & {}^l\hat{\Sigma}_M^{l+1:L,l+1:L} \end{pmatrix} \right). \tag{23}$$

*Here $\hat{\mu}_n^l$ and $\hat{\Sigma}_n^l$ are as in Eqs. (10) and (11), respectively, and we defined*

$${}^l\hat{\mu}_M^{l+1:L} = \mu_M^{l+1:L} + S_M^{l+1:L,1:l-1} \mathrm{diag}(\widetilde{\mathcal{K}}_{Mn}^{1:l-1}) \left( \widetilde{S}_n^{1:l-1,1:l-1} \right)^{-1} (f_n^{1:l-1} - \widetilde{\mu}_n^{1:l-1})$$

$${}^l\hat{\Sigma}_M^{l+1:L,l+1:L} = S_M^{l+1:L,l+1:L} - S_M^{l+1:L,1:l-1} \mathrm{diag}(\widetilde{\mathcal{K}}_{Mn}^{1:l-1}) \left( \widetilde{S}_n^{1:l-1,1:l-1} \right)^{-1} \mathrm{diag}(\widetilde{\mathcal{K}}_{nM}^{1:l-1}) S_M^{1:l-1,l+1:L} \tag{24}$$

$${}^l\hat{\Sigma}_{nM}^{l,l+1:L} = \widetilde{\mathcal{K}}_{nM}^l S_M^{l,l+1:L} - \widetilde{S}_n^{l,1:l-1} \left( \widetilde{S}_n^{1:l-1,1:l-1} \right)^{-1} \mathrm{diag}(\widetilde{\mathcal{K}}_{nM}^{1:l-1}) S_M^{1:l-1,l+1:L}.$$

In the equations above we used $\text{diag}(A^{1:l})$ to denote the formation of a block diagonal matrix, where the diagonal blocks are given by $A^1, \ldots, A^l$. Note that while we only need one index to label $\hat{\mu}_n^l$ and $\hat{\Sigma}_n^l$, we need several for the objects defined in Eq. (24). Take e.g. ${}^l\hat{\Sigma}_M^{l+1:L,l+1:L}$: The upper left index denotes for which $l$ the formula is valid (which will become important when we do the induction step $l \to l+1$). The upper right indices (try to) capture which terms of $S_M$ are most important for the definition, they have nothing to do with the dimensionality of the objects. (In fact, the matrix ${}^l\hat{\Sigma}_M$ contains $L-l-1 \times L-l-1$ blocks of various sizes $T_l M \times T_{l'} M$.) This makes it easier later on when we do calculations with these objects.

Before we prove Lem. 2, we will first show how its results can be used to prove Thm. 1:

**Proof of Theorem 1**. As shown in Appx. D, we can write

$$q(f_n^L) = \int \left[ \int q(f_M) \prod_{l=1}^L p(f_n^l | f_M^l; f_n^{l-1}) df_M \right] df_n^1 \cdots df_n^{L-1}. \tag{25}$$

Obtaining a formula for the inner integral can be done using Lem. 2 with $l = L$, in which case Eq. (22) reads

$$\int q(f_M^1, \ldots, f_M^L) \prod_{l'=1}^L p(f_n^{l'} | f_M^{l'}) df_M^{l'} = \left[ \prod_{l'=1}^{L-1} q(f_n^{l'} | f_n^1, \ldots, f_n^{l'-1}) \right] q(f_n^L | f_n^1, \ldots, f_n^{L-1}) \tag{26}$$

since there is nothing left to integrate over. According to Eqs. (9) and (23) the distribution $q(f_n^L | f_n^1, \ldots, f_n^{L-1})$ has the form necessary to be written as part of the product and we therefore have

$$\int q(f_M^1, \ldots, f_M^L) \prod_{l'=1}^L p(f_n^{l'} | f_M^{l'}) df_M^{l'} = \prod_{l'=1}^L q(f_n^{l'} | f_n^1, \ldots, f_n^{l'-1}). \tag{27}$$

Plugging this into Eq. (25) yields

$$q(f_n^L) = \int \prod_{l'=1}^L q(f_n^{l'} | f_n^1, \ldots, f_n^{l'-1}) df_n^1 \cdots df_n^{L-1}, \tag{28}$$

where the distributions $q$ on the right hand side have the properties described in Eqs. (9) - (11). $\qquad\square$

In order to prove Lem. 2, we will regularly need two standard formulas from Gaussian calculus, namely conditioning multivariate Gaussians,

$$\mathcal{N}\left( \begin{pmatrix} x \\ y \end{pmatrix} \middle| \begin{pmatrix} a \\ b \end{pmatrix}, \begin{pmatrix} A & C \\ C^\top & B \end{pmatrix} \right) = \mathcal{N}(x|a, A) \, \mathcal{N}\left( y | b + C^\top A^{-1}(x-a), B - C^\top A^{-1} C \right), \tag{29}$$

and solving Gaussian integrals ("propagation"):

$$\int \mathcal{N}(x|a + Fy, A) \, \mathcal{N}(y|b, B) \, dy = \mathcal{N}\left( x | a + Fb, A + FBF^\top \right). \tag{30}$$

**Proof of Lemma 2**. As already said, we prove the lemma by induction:

**Base case**  We need to show that Eq. (22) holds for $l = 1$, i.e., that

$$\int q(f_M^1, \ldots, f_M^L) \prod_{l'=1}^L p(f_n^{l'} | f_M^{l'}) df_M^{l'} = \int q(f_n^1, f_M^2, \ldots, f_M^L) \prod_{l'=2}^L p(f_n^{l'} | f_M^{l'}) df_M^{l'}, \tag{31}$$

where $q(f_n^1, f_M^2, \ldots, f_M^L)$ is given according to Eqs. (23) and (24).

In order to do so, we will perform the following steps:

i) Starting with the LHS of Eq. (31), we isolate all terms that depend on $f_M^1$:

$$\int \left[ \int q(f_M^1, \ldots, f_M^L) p(f_n^1 | f_M^1) df_M^1 \right] \prod_{l'=2}^{L} p(f_n^{l'} | f_M^{l'}) df_M^{l'}. \tag{32}$$

ii) In the previous equation, we only consider the inner integral and condition $q$ on $f_M^1$:

$$\int q(f_M^1, \ldots, f_M^L) p(f_n^1 | f_M^1) df_M^1 = \int q(f_M^1) q(f_M^2, \ldots, f_M^L | f_M^1) p(f_n^1 | f_M^1) df_M^1. \tag{33}$$

iii) Next, we obtain the joint distribution of the two terms that are conditioned on $f_M^1$:

$$\int q(f_M^1) q(f_M^2, \ldots, f_M^L | f_M^1) p(f_n^1 | f_M^1) df_M^1 = \int q(f_M^1) q(f_n^1, f_M^2, \ldots, f_M^L | f_M^1) df_M^1. \tag{34}$$

iv) Then we evaluate the integral:

$$\int q(f_M^1) q(f_n^1, f_M^2, \ldots, f_M^L | f_M^1) df_M^1 = q(f_n^1, f_M^2, \ldots, f_M^L). \tag{35}$$

v) Finally, we check that the resulting distribution is given by Eqs. (23) and (24). This then proves the equality in Eq. (31).

Step ii) is the first one where we actually need to calculate something, namely the conditioning of $q(f_M^1, \ldots, f_M^L)$. Using its definition in Eq. (18) and performing the conditioning according to Eq. (29) yields

$$\begin{aligned}
q(f_M^1, \ldots, f_M^L) &= q(f_M^1) q(f_M^2, \ldots, f_M^L | f_M^1) \\
&= \mathcal{N}\left(f_M^1 | \mu_M^1, S_M^{11}\right) \mathcal{N}\left(f_M^{2:L} \middle| \mu_M^{2:L} + S_M^{2:L,1} \left(S_M^{11}\right)^{-1} (f_M^1 - \mu_M^1), S_M^{2:L,2:L} - S_M^{2:L,1} \left(S_M^{11}\right)^{-1} S_M^{1,2:L}\right).
\end{aligned} \tag{36}$$

For step iii) we use the formula we just obtained for $q(f_M^2, \ldots, f_M^L | f_M^1)$ and additionally $p(f_n^1 | f_M^1)$, which, according to Eq. (19) is given by $\mathcal{N}\left(f_n^1 \middle| \widetilde{\mathcal{K}}_{nM}^1 f_M^1, \widetilde{\mathcal{K}}_{nn}^1\right)$, and then proceed to build their joint Gaussian distribution:

$$\begin{aligned}
q(f_n^1, f_M^2, \ldots, f_M^L | f_M^1) &= p(f_n^1 | f_M^1) q(f_M^2, \ldots, f_M^L | f_M^1) \\
&= \mathcal{N}\left(\begin{pmatrix} f_n^1 \\ f_M^{2:L} \end{pmatrix} \middle| \begin{pmatrix} \widetilde{\mathcal{K}}_{nM}^1 f_M^1 \\ \mu_M^{2:L} + S_M^{2:L,1} \left(S_M^{11}\right)^{-1} (f_M^1 - \mu_M^1) \end{pmatrix}, \begin{pmatrix} \widetilde{\mathcal{K}}_{nn}^1 & 0 \\ 0 & S_M^{2:L,2:L} - S_M^{2:L,1} \left(S_M^{11}\right)^{-1} S_M^{1,2:L} \end{pmatrix}\right).
\end{aligned} \tag{37}$$

In step iv) we perform the integration using the term above for the joint and $q(f_M^1) = \mathcal{N}\left(f_M^1 | \mu_M^1, S_M^{11}\right)$ from Eq. (36) for the marginal. Applying Eq. (30) yields

$$\begin{aligned}
&\int q(f_n^1, f_M^2, \ldots, f_M^L | f_M^1) q(f_M^1) df_M^1 \\
&= \mathcal{N}\left(\begin{pmatrix} f_n^1 \\ f_M^{2:L} \end{pmatrix} \middle| \begin{pmatrix} \widetilde{\mathcal{K}}_{nM}^1 \mu_M^1 \\ \mu_M^{2:L} + S_M^{2:L,1} \left(S_M^{11}\right)^{-1} (\mu_M^1 - \mu_M^1) \end{pmatrix}, \right. \\
&\quad \left. \begin{pmatrix} \widetilde{\mathcal{K}}_{nn}^1 & 0 \\ 0 & S_M^{2:L,2:L} - S_M^{2:L,1} \left(S_M^{11}\right)^{-1} S_M^{1,2:L} \end{pmatrix} + \begin{pmatrix} \widetilde{\mathcal{K}}_{nM}^1 \\ S_M^{2:L,1} \left(S_M^{11}\right)^{-1} \end{pmatrix} S_M^{11} \begin{pmatrix} \widetilde{\mathcal{K}}_{nM}^1 \\ S_M^{2:L,1} \left(S_M^{11}\right)^{-1} \end{pmatrix}^{\top}\right) \\
&= \mathcal{N}\left(\begin{pmatrix} f_n^1 \\ f_M^{2:L} \end{pmatrix} \middle| \begin{pmatrix} \widetilde{\mu}_n^1 \\ \mu_M^{2:L} \end{pmatrix}, \begin{pmatrix} \widetilde{S}_n^{11} & \widetilde{\mathcal{K}}_{nM}^1 S_M^{1,2:L} \\ S_M^{2:L,1} \widetilde{\mathcal{K}}_{Mn}^1 & S_M^{2:L,2:L} \end{pmatrix}\right).
\end{aligned} \tag{38}$$

In order to arrive at the last line we simplified the terms and used the definitions of $\widetilde{\mu}_n^1$ and $\widetilde{S}_n^{11}$ in Thm. 1.

Step v) requires us to evaluate Eq. (23) for $l = 1$ resulting in

$$\mathcal{N}\left(\begin{pmatrix} f_n^1 \\ f_M^{2:L} \end{pmatrix} \middle| \begin{pmatrix} \hat{\mu}_n^1 \\ {}^1\hat{\mu}_M^{2:L} \end{pmatrix}, \begin{pmatrix} \hat{\Sigma}_n^1 & {}^1\hat{\Sigma}_{nM}^{1,2:L} \\ \left({}^1\hat{\Sigma}_{nM}^{l,2:L}\right)^\top & {}^1\hat{\Sigma}_M^{2:L,2:L} \end{pmatrix}\right), \tag{39}$$

which is the term $q(f_n^1, f_M^2, \ldots, f_M^L)$ on the RHS of Eq. (31). Plugging in the definitions from Eq. (24) we can easily verify that this last term indeed agrees with Eq. (38). Therefore our statement in Lem. 2 holds for $l = 1$.

**Inductive step** We assume that Lemma 2 holds for some $l = 1, \ldots, L-1$ (induction hypothesis) and then need to show that it also holds for $l+1$. That is, assuming that

$$\int q(f_M^{1:L}) \prod_{l'=1}^{L} p(f_n^{l'}|f_M^{l'}) df_M^{l'} = \left[\prod_{l'=1}^{l-1} q(f_n^{l'}|f_n^{1:l'-1})\right] \int q(f_n^l, f_M^{l+1:L}|f_n^{1:l-1}) \prod_{l'=l+1}^{L} p(f_n^{l'}|f_M^{l'}) df_M^{l'}, \tag{40}$$

holds for some $l$ with the terms on the RHS given by Eqs. (23), and (24) we need to show that we can also write the previous equation as

$$\left[\prod_{l'=1}^{l} q(f_n^{l'}|f_n^{1:l'-1})\right] \int q(f_n^{l+1}, f_M^{l+2:L}|f_n^{1:l}) \prod_{l'=l+2}^{L} p(f_n^{l'}|f_M^{l'}) df_M^{l'}, \tag{41}$$

where this time the terms are given by Eqs. (23), and (24) but with $l \to l+1$.

The way to show this is very similar to the way we showed the base case, the resulting formulas will only look more complicated and we will need one additional step in the beginning:

o) Assuming that Eq. (40) holds for some $l$, we can start immediately with the RHS. The first step will be to marginalise $f_n^l$ from the distribution $q$ within the integral and show that the resulting marginal $q(f_n^l|f_n^{1:l-1})$ has the right form to be written as part of the product in front of the integral:

$$\left[\prod_{l'=1}^{l-1} q(f_n^{l'}|f_n^{1:l'-1})\right] \int q(f_n^l, f_M^{l+1:L}|f_n^{1:l-1}) \prod_{l'=l+1}^{L} p(f_n^{l'}|f_M^{l'}) df_M^{l'} \tag{42}$$

$$= \left[\prod_{l'=1}^{l-1} q(f_n^{l'}|f_n^{1:l'-1})\right] \int q(f_n^l|f_n^{1:l-1}) q(f_M^{l+1:L}|f_n^{1:l}) \prod_{l'=l+1}^{L} p(f_n^{l'}|f_M^{l'}) df_M^{l'} \tag{43}$$

$$= \left[\prod_{l'=1}^{l} q(f_n^{l'}|f_n^{1:l'-1})\right] \int q(f_M^{l+1:L}|f_n^{1:l}) \prod_{l'=l+1}^{L} p(f_n^{l'}|f_M^{l'}) df_M^{l'}. \tag{44}$$

Having done this, we will have to do the exact same steps as in the base case, which we will repeat below with updated indices.

i) Continuing from Eq. (44), we isolate all terms that depend on $f_M^{l+1}$:

$$\left[\prod_{l'=1}^{l} q(f_n^{l'}|f_n^{1:l'-1})\right] \int \left[\int q(f_M^{l+1:L}|f_n^{1:l}) p(f_n^{l+1}|f_M^{l+1}) df_M^{l+1}\right] \prod_{l'=l+2}^{L} p(f_n^{l'}|f_M^{l'}) df_M^{l'}. \tag{45}$$

ii) Comparing this to Eq. (41), we see that it remains to be shown that the inner integral equals $q(f_n^{l+1}, f_M^{l+2:L}|f_n^{1:l})$ [given by Eqs. (23) and (24)]. Therefore we only consider the inner integral and therein condition $q$ on $f_M^{l+1}$:

$$\int q(f_M^{l+1:L}|f_n^{1:l}) p(f_n^{l+1}|f_M^{l+1}) df_M^{l+1} = \int q(f_M^{l+1}|f_n^{1:l}) q(f_M^{l+2:L}|f_n^{1:l}, f_M^{l+1}) p(f_n^{l+1}|f_M^{l+1}) df_M^{l+1}. \tag{46}$$

iii) Next, we obtain the joint distribution of the two terms that are conditioned on $f_M^{l+1}$:

$$\int q(f_M^{l+1}|f_n^{1:l}) q(f_M^{l+2:L}|f_n^{1:l}, f_M^{l+1}) p(f_n^{l+1}|f_M^{l+1}) df_M^{l+1} = \int q(f_M^{l+1}|f_n^{1:l}) q(f_n^{l+1}, f_M^{l+2:L}|f_n^{1:l}, f_M^{l+1}) df_M^{l+1}. \tag{47}$$

iv) Then we evaluate the integral:

$$\int q(f_M^{l+1}|f_n^{1:l})q(f_n^{l+1}, f_M^{l+2:L}|f_n^{1:l}, f_M^{l+1})df_M^{l+1} = q(f_n^{l+1}, f_M^{l+2:L}|f_n^{1:l}). \tag{48}$$

v) Finally, we check that the resulting distribution is given by Eqs. (23) and (24). This then proves the equality of Eqs. (40) and (41).

Let us begin with step o): According to Eq. (23), we have

$$q(f_n^l, f_M^{l+1:L}|f_n^{1:l-1}) = \mathcal{N}\left(\begin{pmatrix} f_n^l \\ f_M^{l+1:L} \end{pmatrix}\middle|\begin{pmatrix} \hat{\mu}_n^l \\ {}^l\hat{\mu}_M^{l+1:L} \end{pmatrix}, \begin{pmatrix} \hat{\Sigma}_n^l & {}^l\hat{\Sigma}_{nM}^{l,l+1:L} \\ \left({}^l\hat{\Sigma}_{nM}^{l,l+1:L}\right)^\top & {}^l\hat{\Sigma}_M^{l+1:L,l+1:L} \end{pmatrix}\right), \tag{49}$$

which we condition on $f_n^l$ using Eq. (29) (i.e., going from Eq. (42) to Eq. (43)):

$$q(f_n^l, f_M^{l+1:L}|f_n^{1:l-1}) = q(f_n^l|f_n^{1:l-1})q(f_M^{l+1:L}|f_n^{1:l}) = \mathcal{N}\left(f_n^l\middle|\hat{\mu}_n^l, \hat{\Sigma}_n^l\right) \times$$

$$\mathcal{N}\left(f_M^{l+1:L}\middle|{}^l\hat{\mu}_M^{l+1:L} + \left({}^l\hat{\Sigma}_{nM}^{l+1:L,l}\right)^\top\left(\hat{\Sigma}_n^l\right)^{-1}(f_n^l - \hat{\mu}_n^l), {}^l\hat{\Sigma}_M^{l+1:L,l+1:L} - \left({}^l\hat{\Sigma}_{nM}^{l+1:L,l}\right)^\top\left(\hat{\Sigma}_n^l\right)^{-1}{}^l\hat{\Sigma}_{nM}^{l,l+1:L}\right). \tag{50}$$

We therefore see that $q(f_n^l|f_n^{1:l-1}) = \mathcal{N}\left(f_n^l\middle|\hat{\mu}_n^l, \hat{\Sigma}_n^l\right)$, which is the right form for it to be included in the product in front of the integral in Eq. (43). This lets us arrive at Eq. (44), hence finishing step o).

In step i) nothing really happens, we just note that, according to Eq. (19),

$$p(f_n^{l+1}|f_M^{l+1}) = \mathcal{N}\left(f_n^{l+1}\middle|\widetilde{\mathcal{K}}_{nM}^{l+1}f_M^l, \widetilde{\mathcal{K}}_{nn}^{l+1}\right). \tag{51}$$

Using $q(f_M^{l+1:L}|f_n^{1:l})$ from Eq. (50), we perform step ii) according to Eq. (29), resulting in

$$q(f_M^{l+1:L}|f_n^{1:l}) = q(f_M^{l+1}|f_n^{1:l})q(f_M^{l+2:L}|f_n^{1:l}, f_M^{l+1}), \tag{52}$$

where

$$q(f_M^{l+1}|f_n^{1:l}) = \mathcal{N}\left(f_M^{l+1}\middle|{}^l\hat{\mu}_M^{l+1} + \left({}^l\hat{\Sigma}_{nM}^{l+1,l}\right)^\top\left(\hat{\Sigma}_n^l\right)^{-1}(f_n^l - \hat{\mu}_n^l), {}^l\hat{\Sigma}_M^{l+1,l+1} - \left({}^l\hat{\Sigma}_{nM}^{l+1,l}\right)^\top\left(\hat{\Sigma}_n^l\right)^{-1}{}^l\hat{\Sigma}_{nM}^{l,l+1}\right) \tag{53}$$

and

$$q(f_M^{l+2:L}|f_n^{1:l}, f_M^{l+1})$$
$$= \mathcal{N}\left(f_M^{l+2:L}\middle|{}^l\hat{\mu}_M^{l+2:L} + \left({}^l\hat{\Sigma}_{nM}^{l+2:L,l}\right)^\top\left(\hat{\Sigma}_n^l\right)^{-1}(f_n^l - \hat{\mu}_n^l) + \left({}^l\hat{\Sigma}_M^{l+2:L,l+1} - \left({}^l\hat{\Sigma}_{nM}^{l+2:L,l}\right)^\top\left(\hat{\Sigma}_n^l\right)^{-1}{}^l\hat{\Sigma}_{nM}^{l,l+1}\right) \times\right.$$
$$\left({}^l\hat{\Sigma}_M^{l+1,l+1} - \left({}^l\hat{\Sigma}_{nM}^{l+1,l}\right)^\top\left(\hat{\Sigma}_n^l\right)^{-1}{}^l\hat{\Sigma}_{nM}^{l,l+1}\right)^{-1}\left(f_M^{l+1} - {}^l\hat{\mu}_M^{l+1} - \left({}^l\hat{\Sigma}_{nM}^{l+1,l}\right)^\top\left(\hat{\Sigma}_n^l\right)^{-1}(f_n^l - \hat{\mu}_n^l)\right),$$
$${}^l\hat{\Sigma}_M^{l+2:L,l+2:L} - \left({}^l\hat{\Sigma}_{nM}^{l+2:L,l}\right)^\top\left(\hat{\Sigma}_n^l\right)^{-1}{}^l\hat{\Sigma}_{nM}^{l,l+2:L} - \left({}^l\hat{\Sigma}_M^{l+2:L,l+1} - \left({}^l\hat{\Sigma}_{nM}^{l+2:L,l}\right)^\top\left(\hat{\Sigma}_n^l\right)^{-1}{}^l\hat{\Sigma}_{nM}^{l,l+1}\right) \times$$
$$\left({}^l\hat{\Sigma}_M^{l+1,l+1} - \left({}^l\hat{\Sigma}_{nM}^{l+1,l}\right)^\top\left(\hat{\Sigma}_n^l\right)^{-1}{}^l\hat{\Sigma}_{nM}^{l,l+1}\right)^{-1}\left({}^l\hat{\Sigma}_M^{l+1,l+2:L} - \left({}^l\hat{\Sigma}_{nM}^{l+1,l}\right)^\top\left(\hat{\Sigma}_n^l\right)^{-1}{}^l\hat{\Sigma}_{nM}^{l,l+2:L}\right)\right). \tag{54}$$

For step iii) we have to build the joint Gaussian distribution

$$q(f_M^{l+2:L}|f_n^{1:l}, f_M^{l+1})p(f_n^{l+1}|f_M^{l+1}) = q(f_n^{l+1}, f_M^{l+2:L}|f_n^{1:l}, f_M^{l+1}) \tag{55}$$

using Eqs. (51) and (54). Since this formula would be even longer than the one in Eq. (54), we refrain from explicitly writing it here. While the corresponding formula for the base case [Eq. (37)] is much simpler the resulting form of Eq. (55) would be similar.

Next, the integration in step iv) can be performed using Eqs. (30), (53), and (55). The calculations are again very similar to the ones in the corresponding step for the base case [Eq. (38)] so we only state the final result here:

$$q(f_n^{l+1}, f_M^{l+2:L}|f_n^{1:l}) = \int q(f_M^{l+1}|f_n^{1:l})q(f_n^{l+1}, f_M^{l+2:L}|f_n^{1:l}, f_M^{l+1})df_M^{l+1}$$

$$= \mathcal{N}\left(\begin{pmatrix} f_n^{l+1} \\ f_M^{l+2:L} \end{pmatrix} \middle| \begin{pmatrix} \hat{m}_n^{l+1} \\ \hat{m}_M^{l+2:L} \end{pmatrix}, \begin{pmatrix} \hat{S}_n^{l+1} & \hat{S}_{nM}^{l+1,l+2:L} \\ \left(\hat{S}_{nM}^{l+1,l+2:L}\right)^\top & \hat{S}_M^{l+2:L,l+2:L} \end{pmatrix}\right), \tag{56}$$

where

$$\hat{m}_n^{l+1} = \widetilde{\mathcal{K}}_{nM}^{l+1}\left({}^l\hat{\mu}_M^{l+1} + \left({}^l\hat{\Sigma}_{nM}^{l+1,l}\right)^\top \left(\hat{\Sigma}_n^l\right)^{-1}(f_n^l - \hat{\mu}_n^l)\right) \tag{57}$$

$$\hat{m}_M^{l+2:L} = {}^l\hat{\mu}_M^{l+2:L} + \left({}^l\hat{\Sigma}_{nM}^{l+2:L,l}\right)^\top \left(\hat{\Sigma}_n^l\right)^{-1}(f_n^l - \hat{\mu}_n^l) \tag{58}$$

$$\hat{S}_n^{l+1} = \mathcal{K}_{nn}^{l+1} + \mathcal{K}_{nM}^{l+1}\left({}^l\hat{\Sigma}_M^{l+1,l+1} - \left({}^l\hat{\Sigma}_{nM}^{l+1,l}\right)^\top \left(\hat{\Sigma}_n^l\right)^{-1}{}^l\hat{\Sigma}_{nM}^{l,l+1}\right)\mathcal{K}_{Mn}^{l+1} \tag{59}$$

$$\hat{S}_{nM}^{l+1,l+2:L} = \widetilde{\mathcal{K}}_{nM}^{l+1}\left({}^l\hat{\Sigma}_M^{l+1,l+2:L} - \left({}^l\hat{\Sigma}_{nM}^{l+1,l}\right)^\top \left(\hat{\Sigma}_n^l\right)^{-1}{}^l\hat{\Sigma}_{nM}^{l,l+2:L}\right) \tag{60}$$

$$\hat{S}_M^{l+2:L,l+2:L} = {}^l\hat{\Sigma}_M^{l+2:L,l+2:L} - \left({}^l\hat{\Sigma}_{nM}^{l+2:L,l}\right)^\top \left(\hat{\Sigma}_n^l\right)^{-1}{}^l\hat{\Sigma}_{nM}^{l,l+2:L}. \tag{61}$$

What remains to be shown in step v) is that this result does in fact agree with the expected result from Lem. 2, i.e.,

$$q(f_n^{l+1}, f_M^{l+2:L}|f_n^{1:l}) = \mathcal{N}\left(\begin{pmatrix} f_n^{l+1} \\ f_M^{l+2:L} \end{pmatrix} \middle| \begin{pmatrix} \hat{\mu}_n^{l+1} \\ {}^{l+1}\hat{\mu}_M^{l+2:L} \end{pmatrix}, \begin{pmatrix} \hat{\Sigma}_n^{l+1} & {}^{l+1}\hat{\Sigma}_{nM}^{l+1,l+2:L} \\ \left({}^{l+1}\hat{\Sigma}_{nM}^{l+1,l+2:L}\right)^\top & {}^{l+1}\hat{\Sigma}_M^{l+2:L,l+2:L} \end{pmatrix}\right), \tag{62}$$

where the terms are defined in Eqs. (10), (11), and (24). That means we have to prove that $\hat{m}_n^{l+1} = \hat{\mu}_n^{l+1}$ and similarly for the other terms in Eqs. (58) - (61). Note that this is the point where we need the left indices in order to distinguish e.g. the term ${}^l\hat{\mu}_M^{l+2:L}$ appearing in Eq. (58) from ${}^{l+1}\hat{\mu}_M^{l+2:L}$ appearing in the mean of Eq. (62).

We will exemplarily prove that $\hat{m}_n^{l+1} = \hat{\mu}_n^{l+1}$: Starting from Eq. (57) we have

$$\hat{m}_n^{l+1} = \widetilde{\mathcal{K}}_{nM}^{l+1}\left({}^l\hat{\mu}_M^{l+1} + \left({}^l\hat{\Sigma}_{nM}^{l+1,l}\right)^\top \left(\hat{\Sigma}_n^l\right)^{-1}(f_n^l - \hat{\mu}_n^l)\right)$$

$$= \widetilde{\mu}_n^{l+1} + \widetilde{S}_n^{l+1,1:l-1}\left(\widetilde{S}_n^{1:l-1,1:l-1}\right)^{-1}(f_n^{1:l-1} - \widetilde{\mu}_n^{1:l-1}) + \left(\widetilde{S}_n^{l+1,l} - \widetilde{S}_n^{l+1,1:l-1}\left(\widetilde{S}_n^{1:l-1,1:l-1}\right)^{-1}\widetilde{S}_n^{1:l-1,l}\right) \times$$

$$\left(\hat{\Sigma}_n^l\right)^{-1}\left(f_n^l - \widetilde{\mu}_n^l - \widetilde{S}_n^{l,1:l-1}\left(\widetilde{S}_n^{1:l-1,1:l-1}\right)^{-1}(f_n^{1:l-1} - \widetilde{\mu}_n^{1:l-1})\right), \tag{63}$$

where we used the definitions in Eqs. (10) and (24) for the terms $\hat{\cdot}$. Note that these definitions are part of the induction hypothesis. It will soon become clear why we did not substitute $\hat{\Sigma}_n^l$. We furthermore used the definitions of the $\widetilde{\mu}_n$ and $\widetilde{S}_n$ terms in Thm. 1 to absorb the $\widetilde{\mathcal{K}}$ terms. In the following we are going to write Eq. (63) in a vectorized form and additionally substitute

$$A = \widetilde{S}_n^{1:l-1,1:l-1}, \qquad B = \widetilde{S}_n^{1:l-1,l}, \qquad C = \widetilde{S}_n^{l,1:l-1}, \qquad \widetilde{D} = \hat{\Sigma}_n^l, \tag{64}$$

The reason for these steps will become clear after two more equations:

$$\hat{m}_n^{l+1} = \widetilde{\mu}_n^{l+1} + \begin{pmatrix} \widetilde{S}_n^{l+1,1:l-1}A^{-1} - \left(\widetilde{S}_n^{l+1,l} - \widetilde{S}_n^{l+1,1:l-1}A^{-1}B\right)\widetilde{D}^{-1}CA^{-1} \\ \left(\widetilde{S}_n^{l+1,l} - \widetilde{S}_n^{l+1,1:l-1}A^{-1}B\right)\widetilde{D}^{-1} \end{pmatrix}^\top \begin{pmatrix} f_n^{1:l-1} - \widetilde{\mu}_n^{1:l-1} \\ f_n^l - \widetilde{\mu}_n^l \end{pmatrix} \tag{65}$$

Going one step further, we recognize this as a vector matrix multiplication,

$$\hat{m}_n^{l+1} = \widetilde{\mu}_n^{l+1} + \begin{pmatrix} \widetilde{S}_n^{l+1,1:l-1} \\ \widetilde{S}_n^{l+1,l} \end{pmatrix}^\top \begin{pmatrix} A^{-1} + A^{-1}B\widetilde{D}^{-1}CA^{-1} & -A^{-1}B\widetilde{D}^{-1} \\ -\widetilde{D}^{-1}CA^{-1} & \widetilde{D}^{-1} \end{pmatrix} \begin{pmatrix} f_n^{1:l-1} - \widetilde{\mu}_n^{1:l-1} \\ f_n^l - \widetilde{\mu}_n^l \end{pmatrix}, \tag{66}$$

where we additionally exploited that $A$ and $\widetilde{D}$ are symmetric and that $B^\top = C$. In order to get any further from here we need the block matrix inversion lemma, which states that

$$
\begin{pmatrix} A & B \\ C & D \end{pmatrix}^{-1} = \begin{pmatrix} A^{-1} + A^{-1}B\widetilde{D}^{-1}CA^{-1} & -A^{-1}B\widetilde{D}^{-1} \\ -\widetilde{D}^{-1}CA^{-1} & \widetilde{D}^{-1} \end{pmatrix},
\tag{67}
$$

where $\widetilde{D} = D - CA^{-1}B$. Comparing Eqs. (66) and (67) explains why we insisted on vectorising the last few formulas and also our definitions in Eq. (64). Finally, since $\hat{\Sigma}_n^l = \widetilde{S}_n^{ll} - \widetilde{S}_n^{l,1:l-1}\left(\widetilde{S}_n^{1:l-1,1:l-1}\right)^{-1}\widetilde{S}_n^{1:l-1,l}$ [Eq. (11)], we also identify $\widetilde{S}_n^{ll} = D$. We can therefore rewrite Eq. (66) by reversing the block matrix inversion and resubstituting the terms in Eq. (64):

$$
\hat{m}_n^{l+1} = \widetilde{\mu}_n^{l+1} + \begin{pmatrix} \widetilde{S}_n^{l+1,1:l-1} \\ \widetilde{S}_n^{l+1,l} \end{pmatrix}^\top \begin{pmatrix} \widetilde{S}_n^{1:l-1,1:l-1} & \widetilde{S}_n^{1:l-1,l} \\ \widetilde{S}_n^{l,1:l-1} & \widetilde{S}_n^{ll} \end{pmatrix}^{-1} \begin{pmatrix} f_n^{1:l-1} - \widetilde{\mu}_n^{1:l-1} \\ f_n^l - \widetilde{\mu}_n^l \end{pmatrix}
$$

$$
= \widetilde{\mu}_n^{l+1} + \widetilde{S}_n^{l+1,1:l}\left(\widetilde{S}_n^{1:l,1:l}\right)^{-1}\left(f_n^{1:l} - \widetilde{\mu}_n^{1:l}\right).
\tag{68}
$$

In the last step we simply rewrote the vectors and the matrix according to the way we defined the submatrix notation. Comparing the final result to Eq. (10), we realize that this is indeed $\hat{\mu}_n^{l+1}$, i.e., the mean term where we substituted $l \to l+1$. In exactly same way, i.e., by reversing the matrix inversion, we can show that the other parameters of the distribution in Eq. (56) indeed coincide with the respective parameters of the distribution in Eq. (62). Since this was the last part that remained to be shown, we finished the proof of Lem. 2. $\qquad\square$

# B   Intuition for the proof of Theorem 1

In this section, we provide some intuition that might be helpful in understanding parts of the proof of Thm. 1. In the first part, we present a different, heuristic way of obtaining the same results, making use of a mathematically wrong (or at least not mathematically rigorous) step. This helped us come up with the exact form of the theorem that we proved above. The second part gives some intuition on how the formulas appearing in Thm. 1, especially Eqs. (10) and (11), can be interpreted. More precisely, we show how these reduce to the mean-field equations when we plug in the mean-field covariance matrix.

## B.1   Heuristic argument for Theorem 1

The aim of this section is to provide a heuristic argument for Thm. 1 as opposed to the complete proof given in Appx. A. For convenience we recap the starting point of that section: We need to show that

$$
\int q(f_M) \prod_{l=1}^{L} p(f_n^l | f_M^l; f_n^{l-1}) df_M = \prod_{l=1}^{L} q(f_n^l | f_n^1, \dots, f_n^{l-1}),
\tag{69}
$$

where the distributions $q$ on the right hand side are defined in Eqs. (9) - (11). The terms appearing on the left hand side are given by $q(f_M) = \mathcal{N}(f_M | \mu_M, S_M)$, which is interchangeably also denoted as

$$
q(f_M^{1:L}) = \mathcal{N}\left(f_M^{1:L} \middle| \mu_M^{1:L}, S_M^{1:L,1:L}\right) = q\begin{pmatrix} f_M^1 \\ \vdots \\ f_M^L \end{pmatrix} = \mathcal{N}\left(\begin{pmatrix} f_M^1 \\ \vdots \\ f_M^L \end{pmatrix} \middle| \begin{pmatrix} \mu_M^1 \\ \vdots \\ \mu_M^L \end{pmatrix}, \begin{pmatrix} S_M^{11} & \cdots & S_M^{1L} \\ \vdots & \ddots & \vdots \\ S_M^{L1} & \cdots & S_M^{LL} \end{pmatrix}\right),
\tag{70}
$$

and

$$
p(f_n^l | f_M^l; f_n^{l-1}) = \mathcal{N}\left(f_n^l \middle| \widetilde{\mathcal{K}}_{nM}^l f_M^l, \widetilde{\mathcal{K}}_{nn}^l\right),
\tag{71}
$$

where

$$
\widetilde{\mathcal{K}}_{nM}^l = \mathcal{K}_{nM}^l \left(\mathcal{K}_{MM}^l\right)^{-1}
\tag{72}
$$

$$
\widetilde{\mathcal{K}}_{nn}^l = \mathcal{K}_{nn}^l - \mathcal{K}_{nM}^l \left(\mathcal{K}_{MM}^l\right)^{-1} \mathcal{K}_{Mn}^l.
\tag{73}
$$

Let us consider $\prod_{l=1}^{L} p(f_n^l | f_M^l; f_n^{l-1})$ on the left hand side of Eq. (69) as a joint multivariate distribution,

$$p(f_n | f_M) = \mathcal{N}\left(\begin{pmatrix} f_n^1 \\ \vdots \\ f_n^L \end{pmatrix} \middle| \begin{pmatrix} \widetilde{\mathcal{K}}_{nM}^1 & 0 & 0 \\ 0 & \ddots & 0 \\ 0 & 0 & \widetilde{\mathcal{K}}_{nM}^L \end{pmatrix} \begin{pmatrix} f_M^1 \\ \vdots \\ f_M^L \end{pmatrix}, \begin{pmatrix} \widetilde{\mathcal{K}}_{nn}^1 & 0 & 0 \\ 0 & \ddots & 0 \\ 0 & 0 & \widetilde{\mathcal{K}}_{nn}^L \end{pmatrix}\right). \tag{74}$$

Note that this is the point where this "proof" becomes heuristic: The object on the right hand side of Eq. (74) is not really a probability distribution, as the variables over which the distribution is defined (the $f_n^l$) appear as parameters of the distribution itself (as inputs of the covariance matrices, e.g. $\widetilde{\mathcal{K}}_{nn}^{l+1}$ ). In the following we will pretend that rules for (multivariate Gaussian) distributions still apply to this object, making the rest of this proof mathematically wrong. We hope that it can still provide some intuition. Using the standard formula for solving Gaussian integrals ("propagation"),

$$\int \mathcal{N}\left(x | a + Fy, A\right) \mathcal{N}\left(y | b, B\right) dy = \mathcal{N}\left(x | a + Fb, A + FBF^\top\right), \tag{75}$$

we can then easily plug in Eq. (74) in the left hand side of Eq. (69), yielding

$$q(f_n) = \int q(f_M) p(f_n | f_M) df_M = \mathcal{N}\left(\begin{pmatrix} f_n^1 \\ \vdots \\ f_n^L \end{pmatrix} \middle| \begin{pmatrix} \widetilde{\mu}_n^1 \\ \vdots \\ \widetilde{\mu}_n^L \end{pmatrix}, \begin{pmatrix} \widetilde{S}_n^{11} & \cdots & \widetilde{S}_n^{1L} \\ \vdots & \ddots & \vdots \\ \widetilde{S}_n^{L1} & \cdots & \widetilde{S}_n^{LL} \end{pmatrix}\right), \tag{76}$$

where

$$\widetilde{\mu}_n^l = \widetilde{\mathcal{K}}_{nM}^l \mu_M^l, \tag{77}$$

$$\widetilde{S}_n^{ll'} = \delta_{ll'} \mathcal{K}_{nn}^l - \widetilde{\mathcal{K}}_{nM}^l \left(\delta_{ll'} \mathcal{K}_{MM}^l - S_M^{ll'}\right) \widetilde{\mathcal{K}}_{Mn}^{l'}. \tag{78}$$

Note that the latter two definitions also appear in Thm. 1. The expression in Eq. (76) has still the same problem as the expression in Eq. (74) in that it is not a valid distribution (since $\widetilde{\mu}_n^l$ and $\widetilde{S}_n^{ll'}$ depend on $f_n^{l-1}$).

Pretending further, that rules for distributions still apply, we can use the standard formula for conditioning multivariate Gaussians,

$$\mathcal{N}\left(\begin{pmatrix} x \\ y \end{pmatrix} \middle| \begin{pmatrix} a \\ b \end{pmatrix}, \begin{pmatrix} A & C \\ C^\top & B \end{pmatrix}\right) = \mathcal{N}\left(x | a, A\right) \mathcal{N}\left(y | b + C^\top A^{-1}(x - a), B - C^\top A^{-1} C\right), \tag{79}$$

to repeatedly condition Eq. (76), resulting in

$$q(f_n) = \prod_{l=1}^{L} q(f_n^l | f_n^1, \ldots, f_n^{l-1}) \quad \text{where} \quad q(f_n^l | f_n^1, \ldots, f_n^{l-1}) = \mathcal{N}\left(f_n^l \middle| \hat{\mu}_n^l, \hat{\Sigma}_n^l\right), \tag{80}$$

and

$$\hat{\mu}_n^l = \widetilde{\mu}_n^l + \widetilde{S}_n^{l,1:l-1} \left(\widetilde{S}_n^{1:l-1,1:l-1}\right)^{-1} (f_n^{1:l-1} - \widetilde{\mu}_n^{1:l-1}), \tag{81}$$

$$\hat{\Sigma}_n^l = \widetilde{S}_n^{ll} - \widetilde{S}_n^{l,1:l-1} \left(\widetilde{S}_n^{1:l-1,1:l-1}\right)^{-1} \widetilde{S}_n^{1:l-1,l}. \tag{82}$$

In Eqs. (81) and (82) the notation $A^{l,1:l'}$ is used to index a submatrix of the variable $A$, e.g. $A^{l,1:l'} = \left(A^{l,1} \cdots A^{l,l'}\right)$.

The final result, Eqs. (80) - (82), is once again a valid distribution and exactly matches the outcome of the mathematically rigorous proof of Thm. 1 in Appx. A. The latter, while being much more complicated, is necessary since the intermediate expressions [Eqs. (74) and (76)] rely on mathematically wrong (or at least dubious) steps.

## B.2 Mean-field as a structured approximation

Here, we verify that when we plug in the mean-field covariance matrix into the formulas appearing in Thm. 1, we recover the mean-field formulas appearing in Sec. 2.2, i.e., that in this case Eq. (9) reduces to Eq. (6). For convenience we repeat the relevant formulas, starting with the mean-field formula for the marginals of the last layer [Eq. (6)]:

$$q(f_n^L) = \int \prod_{l=1}^{L} q(f_n^l; f_n^{l-1}) df_n^1 \cdots df_n^{L-1}, \quad \text{where} \quad q(f_n^l; f_n^{l-1}) = \prod_{t=1}^{T_l} \mathcal{N}\left(f_n^{l,t} \middle| \widetilde{\mu}_n^{l,t}, \widetilde{\Sigma}_n^{l,t}\right), \tag{83}$$

where the means and covariances of the Gaussians in this equation are given by:

$$\widetilde{\mu}_n^{l,t} = \widetilde{K}_{nM}^l \mu_M^{l,t}, \quad \widetilde{\Sigma}_n^{l,t} = K_{nn}^l - \widetilde{K}_{nM}^l \left( K_{MM}^l - S_M^{l,t} \right) \widetilde{K}_{Mn}^l. \tag{84}$$

The formulas for the fully-coupled variant, starting with the marginals of the last layer [Eq. (9)], read,

$$q(f_n^L) = \int \prod_{l=1}^L q(f_n^l | f_n^1, \ldots, f_n^{l-1}) df_n^1 \cdots df_n^{L-1} \quad \text{where} \quad q(f_n^l | f_n^1, \ldots, f_n^{l-1}) = \mathcal{N}\left( f_n^l \Big| \hat{\mu}_n^l, \hat{\Sigma}_n^l \right), \tag{85}$$

where the means and covariances of the Gaussians in this equation are given by:

$$\hat{\mu}_n^l = \widetilde{\mu}_n^l + \widetilde{S}_n^{l,1:l-1} \left( \widetilde{S}_n^{1:l-1,1:l-1} \right)^{-1} (f_n^{1:l-1} - \widetilde{\mu}_n^{1:l-1}), \tag{86}$$

$$\hat{\Sigma}_n^l = \widetilde{S}_n^{ll} - \widetilde{S}_n^{l,1:l-1} \left( \widetilde{S}_n^{1:l-1,1:l-1} \right)^{-1} \widetilde{S}_n^{1:l-1,l}, \tag{87}$$

where

$$\widetilde{\mu}_n^l = \widetilde{\mathcal{K}}_{nM}^l \mu_M^l, \tag{88}$$

$$\widetilde{S}_n^{ll'} = \delta_{ll'} \mathcal{K}_{nn}^l - \widetilde{\mathcal{K}}_{nM}^l \left( \delta_{ll'} \mathcal{K}_{MM}^l - S_M^{ll'} \right) \widetilde{\mathcal{K}}_{Mn}^{l'}. \tag{89}$$

Here we introduced $\mathcal{K}^l = \left( \mathbb{I}_{T_l} \otimes K^l \right)$ as shorthand for the Kronecker product between the identity matrix $\mathbb{I}_{T_l}$ and the covariance matrix $K^l$, and used $\delta$ for the Kronecker delta.

Having all relevant formulas in one place, we can proceed to show that if we plug in the mean field covariance matrix, $S_M = \mathrm{diag}(\{S_M^{ll}\}_{l=1}^L)$, where $S_M^{ll} = \mathrm{diag}(\{S_M^{l,t}\}_{t=1}^{T_l})$ (see also Fig. 1, left), in Eq. (85), we recover Eq. (83): Removing the correlations between the layers by setting $S_M = \mathrm{diag}(\{S_M^{ll}\}_{l=1}^L)$, also implies that $\widetilde{S}_n^{ll'} = 0$ if $l \neq l'$ [Eq. (89)]. Therefore Eqs. (86) and (87) reduce to $\hat{\mu}_n^l = \widetilde{\mu}_n^l$ and $\hat{\Sigma}_n^l = \widetilde{S}_n^{ll}$, respectively. The resulting variational posterior factorises between the layers with $q(f_n^l; f_n^{l-1}) = \mathcal{N}\left( f_n^l \Big| \widetilde{\mu}_n^l, \widetilde{S}_n^{ll} \right)$. Comparing with Eq. (83), we can already see that the means are equal [since $\widetilde{\mu}_n^l = (\widetilde{\mu}_n^{l,1}, \ldots, \widetilde{\mu}_n^{l,T_l})$, cf. Eqs. (84) and (88)]. Removing the correlations within one layer by setting $S_M^{ll} = \mathrm{diag}(\{S_M^{l,t}\}_{t=1}^{T_l})$ renders the covariance matrix $\widetilde{S}_n^{ll}$ block-diagonal. The diagonal blocks are obtained by evaluating Eq. (89) with $S_M^{ll} = \mathrm{diag}(\{S_M^{l,t}\}_{t=1}^{T_l})$. It is easy to see that these diagonal blocks are equal to $\widetilde{\Sigma}_n^{l,t}$ in Eq. (84) and we fully recover the mean-field solution [Eq. (83)].

## C  ELBO

Here we show how to derive the ELBO of the FC DGP, i.e., Eq. (8), which is given by

$$\mathcal{L} = \sum_{n=1}^N \mathbb{E}_{q(f_n^L)} \left[ \log p(y_n | f_n^L) \right] - \mathrm{KL}[q(f_M)|| \prod_{l=1}^L p(f_M^l)]. \tag{90}$$

While this is already done in the supplemental material of Ref. [24], we will do the derivation once again since our notation is different. For convenience we repeat the relevant formulas from the main text, i.e., the general formula for the ELBO [Eq. (3)], the joint DGP prior [Eq. (2)], and the variational family for the DGP [Eq. (4)], which are given by

$$\mathcal{L} = \int q(f_N, f_M) \log \frac{p(y_N, f_N, f_M)}{q(f_N, f_M)} df_N df_M, \tag{91}$$

$$p(y_N, f_N, f_M) = p(y_N | f_N^L) \prod_{l=1}^L p(f_N^l | f_M^l; f_N^{l-1}) p(f_M^l), \tag{92}$$

$$q(f_N, f_M) = q(f_M) \prod_{l=1}^L p(f_N^l | f_M^l; f_N^{l-1}), \tag{93}$$

respectively. By only using the general form of the distributions, and exploiting that we assumed iid noise, i.e., $p(y_N|f_N^L) = \prod_{n=1}^N p(y_n|f_n^L)$, we can get from Eq. (91) to Eq. (90):

$$\mathcal{L} = \int q(f_N, f_M) \log \frac{p(y_N, f_N, f_M)}{q(f_N, f_M)} df_N df_M = \int q(f_N, f_M) \log \frac{p(y_N|f_N^L) \prod_{l=1}^L p(f_M^l)}{q(f_M)} df_N df_M \quad (94)$$

$$= \int q(f_N, f_M) \log p(y_N|f_N^L) df_N df_M + \int q(f_N, f_M) \log \frac{\prod_{l=1}^L p(f_M^l)}{q(f_M)} df_N df_M, \quad (95)$$

$$= \int q(f_N, f_M) \log \prod_{n=1}^N p(y_n|f_n^L) df_N df_M + \int q(f_M) \log \frac{\prod_{l=1}^L p(f_M^l)}{q(f_M)} df_M, \quad (96)$$

$$= \sum_{n=1}^N \int q(f_n^L) \log p(y_n|f_n^L) df_n^L - \mathrm{KL}[q(f_M) || \prod_{l=1}^L p(f_M^l)]. \quad (97)$$

In the last step we introduced $q(f_n^L)$ as simply summarising all remaining terms in the first integral, hence,

$$q(f_n^L) = \int q(f_N, f_M) df_M \prod_{n' \neq n} df_{n'}^L \prod_{l=1}^{L-1} df_N^l. \quad (98)$$

## D  Marginalisation of (most) latent layer outputs

Here we show how to get from the general form of $q(f_n^L)$ given in Eq. (98) to the starting point of our induction proof [Eq. (12)], where all the latent outputs $f_{n'}^l$ are integrated out for all layers $l$ and for all samples $n' \neq n$:

$$q(f_n^L) = \int \left[ \int q(f_M) \prod_{l=1}^L p(f_n^l|f_M^l; f_n^{l-1}) df_M \right] df_n^1 \cdots df_n^{L-1}. \quad (99)$$

While this is already shown in Remark 2 in Ref. [24] (note that the indices there are not correct), we will provide a bit more detail here and we can also nicely point out where the difference in the formulas for $q(f_n^L)$ arises from. For convenience the relevant formulas are repeated below:

$$q(f_N, f_M) = q(f_M) \prod_{l=1}^L p(f_N^l|f_M^l; f_N^{l-1}), \qquad p(f_N^l|f_M^l; f_N^{l-1}) = \mathcal{N}\left( f_N^l \middle| \widetilde{\mathcal{K}}_{NM}^l f_M^l, \widetilde{\mathcal{K}}_{NN}^l \right), \quad (100)$$

where $\widetilde{\mathcal{K}}_{NM}^l = \mathcal{K}_{NM}^l \left( \mathcal{K}_{MM}^l \right)^{-1}$ and $\widetilde{\mathcal{K}}_{NN}^l = \mathcal{K}_{NN}^l - \mathcal{K}_{NM}^l \left( \mathcal{K}_{MM}^l \right)^{-1} \mathcal{K}_{MN}^l$.

We will start by explicitly writing out Eq. (98) and changing the order of integration:

$$q(f_n^L) = \int q(f_M) \left[ \int \prod_{l=1}^L p(f_N^l|f_M^l; f_N^{l-1}) \prod_{n' \neq n} df_n^L \prod_{l=1}^{L-1} df_N^l \right] df_M. \quad (101)$$

In the following we will only be concerned with the inner integral of the previous equation, which can also be written as

$$\int \left( \int p(f_N^L|f_M^L; f_N^{L-1}) \prod_{n' \neq n} df_{n'}^L \right) \prod_{l=1}^{L-1} p(f_N^l|f_M^l; f_N^{l-1}) df_N^l. \quad (102)$$

Here, the inner integral can be solved by exploiting the nice marginalisation property of multivariate Gaussians,

$$\int p(f_N^L|f_M^L; f_N^{L-1}) \prod_{n' \neq n} df_{n'}^L = \int \mathcal{N}\left( f_N^L \middle| \widetilde{\mathcal{K}}_{NM}^L(f_N^{L-1}) f_M^L, \widetilde{\mathcal{K}}_{NN}^L(f_N^{L-1}) \right) \prod_{n' \neq n} df_{n'}^L \quad (103)$$

$$= \mathcal{N}\left( f_n^L \middle| \widetilde{\mathcal{K}}_{nM}^L(f_N^{L-1}) f_M^L, \widetilde{\mathcal{K}}_{nn}^L(f_N^{L-1}) \right), \quad (104)$$

where we explicitly marked the dependence of the $\widetilde{\mathcal{K}}^L$ terms on the outputs $f_N^{L-1}$. While the $\widetilde{\mathcal{K}}_{nM}^l$ and $\widetilde{\mathcal{K}}_{nn}^l$ could in principle still depend on all the outputs of the previous layer, we see from their definitions after Eq. (100) that they in fact only depend on the marginals $f_n^{L-1}$ and that therefore

$$\mathcal{N}\left(f_n^L \Big| \widetilde{\mathcal{K}}_{nM}^L(f_N^{L-1})f_M^L, \widetilde{\mathcal{K}}_{nn}^L(f_N^{L-1})\right) = \mathcal{N}\left(f_n^L \Big| \widetilde{\mathcal{K}}_{nM}^L(f_n^{L-1})f_M^L, \widetilde{\mathcal{K}}_{nn}^L(f_n^{L-1})\right) = p(f_n^L|f_M^L; f_n^{L-1}). \quad (105)$$

Putting the last two equations together results in

$$\int p(f_N^L|f_M^L; f_N^{L-1}) \prod_{n' \neq n} df_{n'}^L = p(f_n^L|f_M^L; f_n^{L-1}). \quad (106)$$

We can continue with integrating out the $f_N^l$ in Eq. (102) in the same fashion (noting at every layer that we can not marginalise out $f_n^l$ as those are inputs to kernels), arriving at

$$\int \prod_{l=1}^{L} p(f_N^l|f_M^l; f_N^{l-1}) \prod_{n' \neq n} df_n^L \prod_{l=1}^{L-1} df_N^l = \int \prod_{l=1}^{L} p(f_n^l|f_M^l; f_n^{l-1}) df_n^1 \cdots df_n^{L-1}. \quad (107)$$

Plugging this back into Eq. (101) and changing the order of integration once again, yields

$$q(f_n^L) = \int \left[\int q(f_M) \prod_{l=1}^{L} p(f_n^l|f_M^l; f_n^{l-1}) df_M\right] df_n^1 \cdots df_n^{L-1}, \quad (108)$$

which is exactly Eq. (99), the result that we set out to show.

### D.1 Difference between mean-field and fully-coupled

From the previous equation it is also possible to see why a proof as in Appx. A was not necessary for the MF DGP. This is due to the form of the variational posterior over the inducing outputs, given by

$$q(f_M) = \begin{cases} \mathcal{N}\left(f_M|\mu_M, S_M\right) & \text{for the FC DGP,} \\ \prod_{l=1}^{L} \prod_{t=1}^{T_l} \mathcal{N}\left(f_M^{l,t}\Big|\mu_M^{l,t}, S_M^{l,t}\right) & \text{for the MF DGP.} \end{cases} \quad (109)$$

Using $q(f_M)$ from the MF DGP, which can also be written as $q(f_M) = \prod_{l=1}^{L} q(f_M^l)$, the inner integral in Eq. (108) can be rewritten as the product of $l$ integrals,

$$\int q(f_M) \prod_{l=1}^{L} p(f_n^l|f_M^l; f_n^{l-1}) df_M = \prod_{l=1}^{L} \int q(f_M^l) p(f_n^l|f_M^l; f_n^{l-1}) df_M^l, \quad (110)$$

each being a standard integral in Gaussian calculus and the resulting formulas are given in Eqs. (6) and (9). In contrast, a fully coupled multivariate Gaussian can not be written as such a product, which is why the rather straightforward solution presented above is not possible in our case and the proof in Appx. A is needed.

## E  Linear algebra to speed up the code

In this section we provide some guidance through the linear algebra that is exploited in our code to speed up or vectorise calculations. We will focus only on the most expensive terms, i.e., the off-diagonal covariance term $\widetilde{S}_n^{l,1:l-1}$ and how to deal with $\left(\widetilde{S}_n^{1:l-1,1:l-1}\right)^{-1}$, which are both needed to calculate $\hat{\mu}_n^l$ and $\hat{\Sigma}_n^l$ in Eqs. (111) and (112), respectively. First, we show how to deal with the FC DGP and afterwards how the sparsity of $S_M$ for the STAR DGP can be used. The linear algebra that can be exploited for all the other terms, e.g. the KL-divergence, will be provided along with the code. Our implementation is in GPflow [19] which provides all the functionalities that are necessary to deal with GPs in Tensorflow [1].

Before we start we will repeat some formulas for convenience:

$$\hat{\mu}_n^l = \tilde{\mu}_n^l + \tilde{S}_n^{l,1:l-1} \left( \tilde{S}_n^{1:l-1,1:l-1} \right)^{-1} (f_n^{1:l-1} - \tilde{\mu}_n^{1:l-1}), \tag{111}$$

$$\hat{\Sigma}_n^l = \tilde{S}_n^{ll} - \tilde{S}_n^{l,1:l-1} \left( \tilde{S}_n^{1:l-1,1:l-1} \right)^{-1} \tilde{S}_n^{1:l-1,l}, \tag{112}$$

$$\tilde{S}_n^{ll'} = \left( \widetilde{\mathcal{K}}_{Mn}^l \right)^\top S_M^{ll'} \widetilde{\mathcal{K}}_{Mn}^{l'}, \qquad \text{if } l \neq l'. \tag{113}$$

Additionally, here is a more explicit definition of our notation of the covariance matrix $S_M$:

$$S_M = \begin{pmatrix} S_M^{11} & \cdots & S_M^{1L} \\ \vdots & \ddots & \vdots \\ S_M^{L1} & \cdots & S_M^{LL} \end{pmatrix}, \qquad S_M^{ll'} = \begin{pmatrix} \left( S_M^{ll'} \right)_{11} & \cdots & \left( S_M^{ll'} \right)_{1T_{l'}} \\ \vdots & \ddots & \vdots \\ \left( S_M^{ll'} \right)_{T_l 1} & \cdots & \left( S_M^{ll'} \right)_{T_l T_{l'}} \end{pmatrix}, \tag{114}$$

where $S_M$, $S_M^{ll'}$, and $\left( S_M^{ll'} \right)_{tt'}$ are matrices of size $MT \times MT$ (where $T = \sum_{l=1}^L T_l$), $MT_l \times MT_{l'}$, and $M \times M$, which store the covariances of the inducing outputs between all inducing points, only those between layer $l$ and $l'$, and only those between the $t$-th task in layer $l$ and the $t'$-th task in layer $l'$, respectively. In order to ensure that $S_M$ is a valid covariance matrix (positive definite) we will numerically only work with its Cholesky decomposition $L_S$ (s.t. $S_M = L_S L_S^\top$), which is a lower triangular matrix. Wherever possible we will want to avoid actually computing $S_M$ and instead calculate all quantities from $L_S$ directly.

### E.1 Fully coupled DGP

**Off-diagonal covariance terms**  The terms

$$\tilde{S}_n^{l,1:l-1} = \begin{pmatrix} \tilde{S}_n^{l1} & \tilde{S}_n^{l2} & \cdots & \tilde{S}_n^{l,l-1} \end{pmatrix}, \tag{115}$$

which are of size $T_l \times \sum_{l'=1}^{l-1} T_{l'}$, have to be calculated for $l = 1, \ldots, L$ and for $n = 1, \ldots, N$. As the number of layers $L$ is in practice rather small, we will calculate all the individual matrices $\tilde{S}_n^{ll'}$ in a loop and concatenate them at the end, while we want to avoid a loop over $N$. Using Eq. (113) and that $\widetilde{\mathcal{K}}_{Mn}^l = \mathbb{I}_{T_l} \otimes \widetilde{K}_{Mn}^l$, we see that (for an example with $T_l, T_{l'} = 2$)

$$\tilde{S}_n^{ll'} = \begin{pmatrix} \left( \widetilde{K}_{Mn}^l \right)^\top (S_M^{ll'})_{11} \widetilde{K}_{Mn}^{l'} & \left( \widetilde{K}_{Mn}^l \right)^\top (S_M^{ll'})_{12} \widetilde{K}_{Mn}^{l'} \\ \left( \widetilde{K}_{Mn}^l \right)^\top (S_M^{ll'})_{21} \widetilde{K}_{Mn}^{l'} & \left( \widetilde{K}_{Mn}^l \right)^\top (S_M^{ll'})_{22} \widetilde{K}_{Mn}^{l'} \end{pmatrix}. \tag{116}$$

Writing $\tilde{S}_n^{ll'}$ in this way has two advantages: Firstly, actually performing the multiplication $\widetilde{\mathcal{K}}_{nM}^l S_M^{ll'} \widetilde{\mathcal{K}}_{Mn}^{l'}$ is extremely inefficient as the $\widetilde{\mathcal{K}}_{Mn}^l$ are block diagonal, which we resolved in this formulation. Secondly, we note that exactly the same operation, i.e., multiplying from left and right by $\left( \widetilde{K}_{Mn}^l \right)^\top$ and $\widetilde{K}_{Mn}^{l'}$, respectively, has to be performed on all $T_l T_{l'}$ blocks of size $M \times M$. This can be exploited since tensorflow has an inbuilt batch mode for most of its matrix operations. In the following we will first show how the relevant block $S_M^{ll'}$ can be efficiently obtained and afterwards show how to deal with the batch matrix multiplication for all $n = 1, \ldots, N$.

Let us consider an example with three layers ($L = 3$), the resulting covariance matrix and its Cholesky decomposition:

$$S_M = \begin{pmatrix} S_M^{11} & S_M^{12} & S_M^{13} \\ S_M^{21} & S_M^{22} & S_M^{23} \\ S_M^{31} & S_M^{32} & S_M^{33} \end{pmatrix} = L_S L_S^\top = \begin{pmatrix} L_S^{11} & 0 & 0 \\ L_S^{21} & L_S^{22} & 0 \\ L_S^{31} & L_S^{32} & L_S^{33} \end{pmatrix} \begin{pmatrix} \left( L_S^{11} \right)^\top & \left( L_S^{21} \right)^\top & \left( L_S^{31} \right)^\top \\ 0 & \left( L_S^{22} \right)^\top & \left( L_S^{32} \right)^\top \\ 0 & 0 & \left( L_S^{33} \right)^\top \end{pmatrix}. \tag{117}$$

From this we can read off formulas for the blocks of $S_M$, e.g., $S_M^{32} = \begin{pmatrix} L_S^{31} & L_S^{32} \end{pmatrix} \begin{pmatrix} L_S^{21} & L_S^{22} \end{pmatrix}^\top$, which in general can be written as

$$S_M^{ll'} = L_S^{l,1:l'} \left( L_S^{l',1:l'} \right)^\top, \tag{118}$$

where we exploited that we only need $S_M^{ll'}$ for $l' < l$ (the formula above is not valid for $l' \geq l$). In this way we avoided calculating unnecessary matrix multiplications involving zero blocks.

Avoiding the loop over $N$ requires a bit more linear algebra: For this we note that e.g. the element $\left(\widetilde{S}_n^{ll}\right)_{11}$ can be seen as the $n$-th diagonal element of the $N \times N$ matrix $\left(\widetilde{K}_{MN}^l\right)^\top (S_M^{ll'})_{11} \widetilde{K}_{MN}^{l'}$. Fully calculating this matrix is obviously very inefficient as we only need its diagonal elements. For this we use that, generally, for $q \times p$ matrices $A$, $C^\top$ and $q \times q$ matrices $B$

$$\text{diag}\left(C^\top B A\right) = \text{column\_sum}(C^\top \odot BA), \tag{119}$$

where $\odot$ denotes the elementwise matrix product. The formula can easily be proved by explicitly writing the matrix products as sums and comparing terms on both sides. Using this on all the blocks of $\widetilde{S}_n^{ll'}$ in Eq. (116) in a batched form and reordering the obtained terms afterwards requires some reshaping, which is explained in the code.

The most expensive calculations for this term are obtaining $S_M^{ll'}$ [Eq. (118)], which is $\mathcal{O}(M^3 T_l T_{l'} \sum_{l''=1}^{l'} T_{l''})$ and the multiplication of e.g. $(S_M^{ll'})_{11} \widetilde{K}_{MN}^{l'}$ which has to be done for all $T_l T_{l'}$ blocks of $S_M^{ll'}$ and is therefore $\mathcal{O}(NM^2 T_l T_{l'})$. Both of these operations have to be performed for all $l' = 1, \ldots, l-1$ in Eq. (115) and also for all layers $l = 1, \ldots, L$. The total computational cost of this term is therefore $\mathcal{O}(M^3 \sum_{l=1}^{L} T_l \sum_{l'=1}^{l-1} T_{l'} \sum_{l''=1}^{l'} T_{l''} + NM^2 \sum_{l=1}^{L} T_l \sum_{l'=1}^{l-1} T_{l'})$.

**Dealing with the inverse covariance terms**  First of all, we will never actually calculate $\left(\widetilde{S}_n^{1:l-1,1:l-1}\right)^{-1}$. We will only use (and update) the lower triangular Cholesky decomposition $L_{Sn}^{1:l-1,1:l-1}$ defined by

$$L_{Sn}^{1:l-1,1:l-1} \left(L_{Sn}^{1:l-1,1:l-1}\right)^\top = \widetilde{S}_n^{1:l-1,1:l-1}. \tag{120}$$

This can be done since the inverse term only ever appears in the product $\widetilde{S}_n^{l,1:l-1} \left(\widetilde{S}_n^{1:l-1,1:l-1}\right)^{-1}$, whose transpose can be efficiently obtained via the solution of two triangular systems (which is implemented as cholesky\_solve in tensorflow). For this we obviously need the Cholesky decomposition first. We will point out how this can be efficiently calculated in the following, taking the calculations for the second layer as an example:

After having calculated $\hat{\mu}_n^2$ and $\hat{\Sigma}_n^2$ [Eqs. (111) and (112), respectively] we necessarily still have the quantities $\widetilde{L}_{Sn}^{11}$ (passed on from the first layer calculations), $\widetilde{S}_n^{12}$, and $\widetilde{S}_n^{22}$ in memory. Note that we therefore can completely build the block matrix $\widetilde{S}_n^{1:2,1:2}$ as

$$\widetilde{S}_n^{1:2,1:2} = \begin{pmatrix} \widetilde{L}_{Sn}^{11} \left(\widetilde{L}_{Sn}^{11}\right)^\top & \widetilde{S}_n^{12} \\ \left(\widetilde{S}_n^{12}\right)^\top & \widetilde{S}_n^{22,} \end{pmatrix} \tag{121}$$

from which we could in principle calculate the Cholesky factor directly. A more efficient way is to assume a general block matrix form for the Cholesky factor,

$$\widetilde{L}_{Sn}^{1:2,1:2} = \begin{pmatrix} A & 0 \\ B & C \end{pmatrix}, \tag{122}$$

and then by using Eq. (120) and comparing the terms to Eq. (121) finding formulas for the unknown terms and solve them. These formulas are given by

$$AA^\top = \widetilde{L}_{Sn}^{11} \left(\widetilde{L}_{Sn}^{11}\right)^\top, \tag{123}$$

$$AB^\top = \widetilde{S}_n^{12}, \tag{124}$$

$$BB^\top + CC^\top = \widetilde{S}_n^{22}. \tag{125}$$

From Eq. (123) we recognize $A = \widetilde{L}_{Sn}^{11}$. Next, we can use this and Eq. (124) to find $B^\top$ as the solution of the (triangular) system $\widetilde{L}_{Sn}^{11} B^\top = \widetilde{S}_n^{12}$. The last step is then to obtain $C$ from the Cholesky decomposition of the matrix $\widetilde{S}_n^{22} - BB^\top$, where we used Eq. (125). Note that while we still have to do a Cholesky decomposition, the matrix that

has to be decomposed is (especially for large $l$) considerably smaller than the full matrix $\widetilde{S}_n^{1:2,1:2}$ and the computation therefore much faster. For general $l$ we simply have to substitute $\widetilde{L}_{Sn}^{11} \rightarrow \widetilde{L}_{Sn}^{1:l-1,1:l-1}$, $\widetilde{S}_n^{12} \rightarrow \widetilde{S}_n^{1:l-1,l}$, and $\widetilde{S}_n^{22} \rightarrow \widetilde{S}_n^{ll}$, the rest stays the same. Plugging the obtained results for $A$, $B$, and $C$ in Eq. (122) yields the required "updated" Cholesky factor that needs to be passed on for the calculations in the next layer.

Solving the three equations (123) - (125) for layer $l$ requires $\mathcal{O}(N(T_l^2 \sum_{l'=1}^{l-1} T_{l'} + T_l(\sum_{l'=1}^{l-1} T_{l'})^2 + T_l^3))$ computation time. Doing so for all layers (note that we do not have to do this for the last layer) is therefore $\mathcal{O}(N \sum_{l=1}^{L-1}(T_l^2 \sum_{l'=1}^{l-1} T_{l'} + T_l(\sum_{l'=1}^{l-1} T_{l'})^2 + T_l^3))$

**Computational costs** Since these were the most expensive terms for the FC DGP, the overall computational cost for evaluating the ELBO is therefore

$$
\mathcal{O}\left( N \sum_{l=1}^{L} \left[ M^2 T_l \sum_{l'=1}^{l-1} T_{l'} + T_l^2 \sum_{l'=1}^{l-1} T_{l'} + T_l(\sum_{l'=1}^{l-1} T_{l'})^2 + T_l^3 + \frac{M^3}{N} T_l \sum_{l'=1}^{l-1} T_{l'} \sum_{l''=1}^{l'} T_{l''} \right] \right). \tag{126}
$$

In order to get a better grasp at this we will take an example architecture with $L$ layers and the same number $T_l = \tau$ of GPs per layer except for the last layer in which there is only one GP ($T_L = 1$). In such a case the computational cost simplifies to

$$
\mathcal{O}(NM^2\tau^2 L^2 + N\tau^3 L^3 + M^3\tau^3 L^3), \tag{127}
$$

where we only kept the highest order terms.

### E.2  Stripes-and-Arrow DGP

In the following we will briefly sketch the computational savings that can be achieved for the two terms discussed in the previous section, when the restricted covariance of the STAR DGP is used. Note that in this case the architecture necessarily needs to be the one decribed in the example given in the previous paragraph. The (potential) non-zero $M \times M$ blocks of the covariance matrix can be described by $\left(S_M^{ll'}\right)_{tt'}$, where $t = t'$ (this captures the terms also present in the MF DGP plus the diagonal stripes) or $l = L$ or $l' = L$ (this captures the two sides of the arrowhead). It is easy to see that this form directly translates to the Cholesky decomposition which we will exploit in the next section: The lower diagonal $L_S$ has (potential) non-zero $M \times M$ blocks $\left(L_S^{ll'}\right)_{tt'}$, only if $l \geq l'$ (lower diagonal) and if $t = t'$ (diagonal and stripes) or $l = L$ (arrowhead).

**Computing the covariance matrix from its Cholesky decomposition** We will start by describing how the relevant terms of $S_M$ can be obtained from our chosen sparse representation of $L_S$, where we summarise the non-zero terms in three arrays, $^{\text{diag}}L_S$, $^{\text{stripes}}L_S$, and $^{\text{arrow}}L_S$. The array $^{\text{diag}}L_S$ contains the $\tau(L-1) + 1$ lower diagonal $M \times M$ blocks on the diagonal of $L_S$, where we access the $t$-th block in the $l$-th layer, i.e., $\left(L_S^{ll}\right)_{tt}$, by $^{\text{diag}}L_S^{l,t}$. The array $^{\text{stripes}}L_S$ contains the $\tau \sum_{k=1}^{L-2} k = \frac{\tau}{2}(L-2)(L-1)$ blocks of size $M \times M$, which form the diagonal stripes of $L_S$, where we access $\left(L_S^{ll'}\right)_{tt}$ (where $L > l > l'$) by $^{\text{stripes}}L_S^{l,l',t}$. The remaining $\tau(L-1)$ blocks of size $M \times M$ of the arrowhead are contained in $^{\text{arrow}}L_S$, where we access $\left(L_S^{Ll}\right)_{1t}$ as $^{\text{arrow}}L_S^{l,t}$. See also Fig. 1 for a depiction of the covariance matrix. The different terms of $S_M$ can then be obtained as listed below:

i) Diagonal terms $\left(S_M^{ll}\right)_{tt}$ with $l < L$:

$$
\left(S_M^{ll}\right)_{tt} = {}^{\text{diag}}L_S^{l,t}\left({}^{\text{diag}}L_S^{l,t}\right)^{\top} + \sum_{l'=1}^{l-1} {}^{\text{stripes}}L_S^{l,l',t}\left({}^{\text{stripes}}L_S^{l,l',t}\right)^{\top} \tag{128}
$$

ii) Diagonal term $\left(S_M^{LL}\right)_{11}$:

$$
\left(S_M^{LL}\right)_{11} = {}^{\text{diag}}L_S^{L,1}\left({}^{\text{diag}}L_S^{L,1}\right)^{\top} + \sum_{l=1}^{L-1}\sum_{t=1}^{\tau} {}^{\text{arrow}}L_S^{l,t}\left({}^{\text{arrow}}L_S^{l,t}\right)^{\top} \tag{129}
$$

iii) Stripe terms $\left(S_M^{ll'}\right)_{tt}$ with $L > l > l'$ ($l < l'$ obtained via transposing):

$$\left(S_M^{ll'}\right)_{tt} = {}^{\text{stripes}}L_S^{l,l',t}\left({}^{\text{diag}}L_S^{l',t}\right)^{\top} + \sum_{l''=1}^{l'-1}{}^{\text{stripe}}L_S^{l,l'',t}\left({}^{\text{stripes}}L_S^{l',l'',t}\right)^{\top} \tag{130}$$

iv) Arrow terms $\left(S_M^{Ll}\right)_{1t}$:

$$\left(S_M^{Ll}\right)_{1t} = {}^{\text{arrow}}L_S^{l,t}\left({}^{\text{diag}}L_S^{l,t}\right)^{\top} + \sum_{l'=1}^{l-1}{}^{\text{arrow}}L_S^{l',t}\left({}^{\text{stripes}}L_S^{l,l',t}\right)^{\top} \tag{131}$$

The vectorisation of these equations can be seen in the code. The stripe terms are the most expensive to compute since an individual term has computational cost $\mathcal{O}(M^3 l')$ which has to be done for $l = 2, \ldots, L-1$ and for $l' = 1, \ldots, l-1$ and for all $t = 1, \ldots, \tau$. Therefore calculating all stripe terms is $\mathcal{O}(M^3 L^3 \tau)$ (where we only kept the highest order term).

**Off-diagonal covariance terms**   As before with the FC DGP, the off-diagonal covariance terms $\widetilde{S}_n^{ll'}$ will also be the most expensive to compute. From the general formula in Eq. (113) it is easy to see that $\left(\widetilde{S}_n^{ll'}\right)_{tt'}$ is only non-zero, if the corresponding $\left(S_M^{ll'}\right)_{tt'}$ are non-zero. We showed in the previous paragraph how those can be calculated, so the only step that remains to be done is the equivalent of Eq. (116), where again Eq. (119) can be used.

Doing this for an individual $M \times M$ block can be done in $\mathcal{O}(NM^2)$ time. Since we have to do this for all $\mathcal{O}(\tau L^2)$ blocks, the total computational cost for the off-diagonal covariance terms is $\mathcal{O}(NM^2\tau L^2)$.

**Dealing with the inverse covariance terms**   For calculating (or updating) the Cholesky decomposition of $\widetilde{S}_n^{1:l-1,1:l-1}$ we could in principle use similar ideas as we used above for calculating $S_M$ (since both have the same sparsity pattern). But as we saw in the previous section, this term only incurs computational costs of $\mathcal{O}(N\tau^3 L^3)$ even for the FC DGP, which is for our settings always the least expensive term. We therefore simply reuse the algorithm described in the previous section to deal with this term and live with the resulting computational costs.

**Computational costs**   The total computational costs for calculating the ELBO for the STAR DGP are therefore

$$\mathcal{O}(NM^2\tau L^2 + N\tau^3 L^3 + M^3\tau L^3). \tag{132}$$

# F   Pseudocode

We summarise the algorithm for calculating the ELBO (8) in Algs. 1, 2, and 3 and 4. Alg. 1 shows how the ELBO can be calculated, where we average over multiple samples to reduce noise obtained by sampling through the layers. Alg. 2 zooms into a single layer of the DGP and differentiates the mean-field approach and ours: All coupled DGP approaches compute additional blocks of the covariance matrix $\widetilde{S}_n$ (marked in orange in Alg. 3 for the fully-coupled DGP and in Alg. 4 for the stripes-and-arrow approximation). These blocks lead to a dependency of the output of the current layer $f_n^l$ on its predecessors $f_n^{1:l-1}$ (marked in orange).

---

**Algorithm 1** ELBO for coupled DGP

---

given minibatch $x_{N_b}$, $y_{N_b}$ of size $N_b$, and
number of Monte Carlo repetitions $R$
$\mathcal{L} = 0$
**for** $n = 1 \ldots N_b$ **do**
    list $= [x_n]$                                                   $\triangleright$ accumulates $\widetilde{\mathcal{K}}^l_{nM}, \widetilde{\mu}^l_n, \widetilde{S}^{1:l,1:l}_n, f^l_n$
    **for** $r = 1 \ldots R$ **do**
        **for** $l = 1 \ldots L$ **do**
            $\hat{\mu}^l_n, \hat{\Sigma}^l_n, \text{list} = \text{sample\_layer}(l, \text{list})$                                 $\triangleright$ Alg. 2
            $\mathcal{L} \mathrel{+}= \frac{N}{N_b S} \text{var\_log\_likelihood}(y_n, \hat{\mu}^L_n, \hat{\Sigma}^L_n)$
$\mathcal{L} \mathrel{-}= \text{KL\_term}()$
**return** $\mathcal{L}$                                                         $\triangleright$ ELBO from Eq. (8)

---

**Algorithm 2** sample\_layer$(l, \text{list})$: Return $\hat{\mu}^l_n, \hat{\Sigma}^l_n$ and sample $f^l_n$, update relevant quantities for later layers.

---

                                            $\triangleright$ list contains $\widetilde{\mathcal{K}}^{1:l-1}_{nM}, \widetilde{\mu}^{1:l-1}_n, \widetilde{S}^{1:l-1,1:l-1}_n, f^{1:l-1}_n$
$\widetilde{\mu}^l_n = \text{get\_mu\_tilde}()$                                               $\triangleright$ Definition in Thm. 1
$\widetilde{S}^{l,1:l}_n = \text{get\_S\_tilde}(l, \widetilde{\mathcal{K}}^{1:l}_{nM})$                                        $\triangleright$ Alg. 3
$\hat{\mu}^l_n = \widetilde{\mu}^l_n + \text{correct\_mu}(\text{list}, \widetilde{S}^{l,1:l-1}_n)$                               $\triangleright$ Eq. (10)
$\hat{\Sigma}^l_n = \widetilde{S}^{ll}_n - \text{correct\_Sigma}(\text{list}, \widetilde{S}^{l,1:l-1}_n)$                             $\triangleright$ Eq. (11)
$f^l_n = \text{sample\_multivariate\_gauss}(\hat{\mu}^l_n, \hat{\Sigma}^l_n)$
list $= \text{append}(\text{list}, \widetilde{\mathcal{K}}^l_{nM}, \widetilde{\mu}^l_n, \widetilde{S}^{l,1:l-1}_n, \widetilde{S}^{ll}_n, f^l_n)$
**return** $\hat{\mu}^l_n, \hat{\Sigma}^l_n, \text{list}$                                             $\triangleright$ Return to Alg. 1

---

**Algorithm 3** get\_S\_tilde$(l, \widetilde{\mathcal{K}}^{1:l}_{nM})$: Calculate $\widetilde{S}^{l,1:l-1}_n$ and $\widetilde{S}^{l,l}_n$ according to their definitions in Thm. 1 for the fully-coupled DGP.

---

$(S^{ll'}_M)_{tt'}$ denotes the $M \times M$ block in $S_M$ that contains the covariances of the inducing outputs of the $t$-th GP in the $l$-th layer and the $t'$-th GP in the $l'$-th layer. Analogously for $(\widetilde{S}^{ll'}_n)_{tt'}$.

---

**for** $l' = 1 \ldots l$ **do**
    **for** $t, t' = 1 \ldots T_l, 1 \ldots T_{l'}$ **do**
        **if** $l = l'$ and $t = t'$ **then**
            $(\widetilde{S}^{l,l}_n)_{tt} = K^l_{nn} + \widetilde{K}^l_{nM} \left( (S^{ll}_M)_{tt} - K^l_{MM} \right) \widetilde{K}^l_{Mn}$
        **else**
            $(\widetilde{S}^{l,l'}_n)_{tt'} = \widetilde{K}^l_{nM} (S^{ll'}_M)_{tt'} \widetilde{K}^{l'}_{Mn}$
**return** $\widetilde{S}^{l,1:l}_n$                                                  $\triangleright$ Return to Alg. 2

---

**Algorithm 4** get\_S\_tilde$(l, \widetilde{\mathcal{K}}^{1:l}_{nM})$: Calculate $\widetilde{S}^{l,1:l-1}_n$ and $\widetilde{S}^{l,l}_n$ for the stripes-and-arrow DGP.

---

**for** $l' = 1 \ldots l$ **do**
    **for** $t, t' = 1 \ldots T_l, 1 \ldots T_{l'}$ **do**
        **if** $l = l'$ and $t = t'$ **then**
            $(\widetilde{S}^{l,l}_n)_{tt} = K^l_{nn} + \widetilde{K}^l_{nM} \left( (S^{ll}_M)_{tt} - K^l_{MM} \right) \widetilde{K}^l_{Mn}$
        **else if** $l = L$ or $t = t'$ **then**
            $(\widetilde{S}^{l,l'}_n)_{tt'} = \widetilde{K}^l_{nM} (S^{ll'}_M)_{tt'} \widetilde{K}^{l'}_{Mn}$
**return** $\widetilde{S}^{l,1:l}_n$                                                  $\triangleright$ Return to Alg. 2

---

Table S3: **Extrapolation behaviour on UCI benchmark datasets.** We report marginal test log-likelihoods (the larger, the better) for various methods and various number of layers $L$ in the extrapolation setting. We marked all methods in bold that performed better or as good as SGP in the interpolation and extrapolation scenario, i.e., we simultaneously also looked at Tab. 1 in Sec. 4. We additionally underlined those that are significantly better (non-overlapping confidence intervals) in at least one of the scenarios. Standard errors are obtained by repeating the experiment 10 times.

| Dataset | SGP | SGHMC DGP | | | MF DGP | | STAR DGP | |
|---|---|---|---|---|---|---|---|---|
| (N,D) | L1 | L1 | L2 | L3 | L2 | L3 | L2 | L3 |
| boston (506,13) | -3.49(0.23) | -3.57(0.16) | -3.64(0.11) | -3.64(0.08) | **-3.36(0.17)** | **-3.41(0.19)** | **-3.38(0.18)** | **-3.38(0.18)** |
| energy (768, 8) | -2.90(0.63) | -3.22(0.69) | -3.53(0.89) | -3.26(0.75) | -3.02(0.64) | -3.45(0.86) | -3.08(0.78) | -2.95(0.74) |
| concrete (1030, 8) | -3.91(0.11) | -3.90(0.06) | -4.37(0.19) | -4.71(0.33) | **-3.76(0.10)** | **-3.71(0.09)** | **-3.79(0.08)** | **-3.68(0.08)** |
| wine red (1599,11) | -1.01(0.02) | -1.15(0.02) | -1.22(0.03) | -1.08(0.05) | -1.02(0.02) | -1.01(0.02) | **-1.01(0.02)** | **-1.01(0.02)** |
| kin8nm (8192, 8) | 0.66(0.04) | **0.72(0.03)** | **1.06(0.03)** | **0.94(0.11)** | **0.96(0.03)** | **0.98(0.04)** | **0.98(0.02)** | **0.94(0.03)** |
| power (9568, 4) | -3.44(0.29) | -4.27(0.41) | -4.19(0.38) | -4.09(0.35) | -3.81(0.30) | -3.82(0.31) | -3.95(0.33) | -3.82(0.27) |
| naval (11934,16) | 3.20(0.32) | 2.83(0.09) | 3.16(0.14) | 3.18(0.12) | 2.33(0.43) | 2.26(0.37) | 2.20(0.22) | 2.95(0.27) |
| protein (45730, 9) | -3.20(0.04) | **-3.20(0.03)** | **-3.17(0.02)** | **-3.10(0.02)** | -3.31(0.04) | -3.23(0.06) | -3.23(0.04) | **-3.19(0.05)** |

## G Additional experimental details

In the following, we describe the experimental details necessary to reproduce the results:

- **Data Normalization** Normalization of inputs and outputs to zero mean and unit variance based on the training data.

- **Inducing Inputs** $M = 128$ inducing inputs initialized with k-means.

- **DGP architecture** $L = 3$ hidden layers with $\tau = 5$ latent GPs each and principal components of the training data as mean function.

- **Likelihood** Gaussian likelihood with initial variance $\sigma_0^2 = 0.01$.

- **Kernel** RBF kernel with automatic relevance determination (initial lengthscale $l_0 = 1$, initial variance $\sigma_0^2 = 1$).

- **Optimizer** Adam Optimizer [15] (number of iterations $nIter = 20,000$, mini-batch size $mbs = 512$, number of Monte Carlo for each data points $R = 5$) with exponentially decaying learning rate (learning rate $lr = 0.005$, steps $s = 1000$, rate $r = 0.98$).

- **Early Stopping** In our initial experiments, we experienced overfitting for the variational methods. To prevent this from happening, we used 10% of the training data as a validation set. We performed early-stopping if the performance on the validation set decreased in five successive strips [20]. We did neither use a hold-out dataset for the GP methods, as they have a much smaller number of hyperparameters, nor for the Hamiltonian Monte Carlo approaches, as the correlation between adjacent samples complicates defining a good early-stopping criterion.

- **Comparability** Besides the early stopping criterion, we used the same model architectures, hyperparameter initialisations and optimisation settings across all methods and all datasets.

- **Runtime Assessment** The runtime of the different approximations was assessed for a single gradient update averaged over 2,000 updates on a 6 core i7-8700 CPU.

- **Natural gradients** For some experiments we also employed natural gradients, meaning we trained the model with a mixture of a natural gradient optimiser on all variational parameters and the Adam optimiser on all other parameters, similarly as in Refs. [26, 10]. We additionally used exponentially decaying learning rates (learning rate Adam (natural gradients) $lr = 0.001$ $(0.005)$, steps $s = 1000$, rate $r = 0.99$).

Table S4: **Extrapolation behaviour: direct comparison of DGP methods (part 2).** This table complements Tab. 2. In the fourth and in the last column, the same quantities as in Tab. 2 are shown (see there for a description), where NG marks a method trained with natural gradients and FC stands for fully-coupled. The other columns (2,3, and 5), marked with (dif), show means and standard errors (over 10 repetitions) of the *difference* in marginal test log likelihood averaged over all test points in a single train test split. In each column, we mark numbers in bold if the second method outperforms the first, and in italics if it is the other way around. For the (dif) columns that is the case if the numbers significantly differ from zero. Note that the methods trained with natural gradients perform worse in the extrapolation task than those trained with Adam (as can be seen in the last column), a phenomenon that will have to be looked at more closely in the future.

| Dataset | MF vs. STAR (dif) | SGHMC vs. STAR (dif) | MF NG vs. FC NG | MF NG vs. FC NG (dif) | MF NG vs. MF |
|---|---|---|---|---|---|
| boston | **0.036(0.029)** | **0.27(0.15)** | **0.58(0.04)** | **0.303(0.108)** | **0.63(0.04)** |
| energy | **0.500(0.202)** | **0.31(0.16)** | **0.70(0.05)** | **0.343(0.231)** | **0.77(0.05)** |
| concrete | 0.036(0.060) | **1.03(0.33)** | **0.56(0.02)** | **0.145(0.070)** | **0.61(0.02)** |
| wine red | **0.004(0.003)** | **0.07(0.05)** | **0.57(0.03)** | **0.018(0.003)** | **0.57(0.03)** |
| kin8nm | *-0.040(0.024)* | -0.00(0.11) | **0.59(0.02)** | **0.028(0.016)** | *0.44(0.03)* |
| power | 0.005(0.096) | **0.27(0.11)** | **0.68(0.03)** | **0.355(0.157)** | 0.52(0.05) |
| naval | **0.693(0.527)** | -0.22(0.23) | *0.24(0.07)* | *-0.178(0.112)* | **0.65(0.07)** |
| protein | **0.033(0.014)** | *-0.10(0.03)* | 0.50(0.01) | *-0.016(0.009)* | *0.47(0.02)* |

Figure S5: **ELBO comparison.** We show boxplots of the ELBOs that correspond to the interpolation setting reported in Tab. 1 for all datasets and for 10 random splits, comparing the three layer versions of MF DGP and our STAR DGP. The performance increase of STAR DGP compared to MF DGP is the largest when the dataset is small (*boston, energy, concrete*), while we observed a performance decrease on the dataset *naval*. For the latter, we also observed convergence difficulties in two runs, where the ELBOs are not plotted in the figure (indicated by the arrow).

Table S5: **ELBO comparison for fully-coupled DGP** We report ELBOs (the larger, the better) for the mean-field (MF) and the fully-coupled (FC) method. We used our standard architecture with $M = 128$, $\tau = 5$, and $L = 3$ for both methods and trained them using natural gradients. Standard errors are obtained by repeating the experiment 10 times. We warm-started the optimisation of the fully-coupled method from the converged mean-field solution, using rather small learning rates (learning rate Adam (natural gradients) $lr = 0.001$ $(0.002)$, steps $s = 1000$, rate $r = 0.95$, cf. page 29) for a maximum of 7000 iterations. We marked the significant better performing (non overlapping standard errors) for each dataset in bold. Our structured approximation yields larger ELBOs for all datasets.

| Dataset (N,D) | boston (506,13) | energy (768, 8) | concrete (1030, 8) | wine_red (1599,11) | kin8nm (8192, 8) | power (9568, 4) | naval (11934,16) | protein (45730, 9) |
|---|---|---|---|---|---|---|---|---|
| MF | -510(30) | 510(8) | -910(50) | -1648(8) | -2000(100) | -390(90) | 33000(600) | -44040(140) |
| FC | **-246(5)** | **600(20)** | **-500(10)** | **-1575(4)** | **-1290(70)** | **-10(50)** | **34600(500)** | **-42610(120)** |

Figure S6: **Convergence of SGHMC.** Left: Distribution over MC samples from a randomly chosen inducing output of a DGP with a single layer, equivalent to a SGP. The red line indicates a Normal distribution fitted to the data. Right: For each inducing output, we computed a p-value if its MC samples are normally distributed. The blue line shows the Bonferroni corrected significance threshold $\alpha = 10^{-5}$.

Figure S7: **Extended runtime comparison.** We compare the runtime of our efficient stripes-and-arrow approximation (STAR DGP) versus the fully coupled (FC DGP) and the mean-field approach (MF DGP) on the *protein* UCI dataset. Shown is the runtime of one gradient step in seconds on a logarithmic scale as a function of the number of inducing points $M$, the number of layers $L$ and the number $\tau$ of latent GPs per intermediate layer (from left to right). The dotted grey lines show the theoretical scaling of the runtime of the STAR DGP for the most important term $\mathcal{O}(NM^2\tau L^2)$.

Figure S8: **Additional covariance and precision matrices.** Covariance/precision matrices after optimisation for three different UCI data sets. The first column depicts covariance matrices for our standard architecture, $L = 3, \tau = 5$, while the second column depicts covariance matrices for $L = 4, \tau = 3$. The third column depicts the precision matrices corresponding to the second column, i.e., the inverse matrices. Plotted are natural logarithms of the absolute values of the variational covariance/precision matrices over the inducing outputs.