[Reviews · NeurIPS 2020]

Review 1

Summary and Contributions: While DGP represent a very natural compositional model for deep learning with uncertainty quantification, their use has been somewhat hampered by the significant computational complexity of preforming full posterior inference in these models. Mean-field approximations have addressed some of this computational complexity issue, but as a result lose much of the uncertainty structure of the full posterior. This paper establishes that in principle one can specify a fully coupled variational approximation, and they further demonstrate that despite an increased computational complexity of this method this increase can be balanced by carefully considered choices of the full covariance structure. They introduce an explicit example of a structured covariance matrices – the stripes-and-arrows pattern – and demonstrate that it can balance improved information sharing between layers, with increased computational efficiency on experimental data.

Strengths: The authors first demonstrate theoretically that it is possible to place a fully parameterised Gaussian variational posterior over the global latent layers, leading to a method than can still marginalise over the inducing outputs, but with the potential for better calibrated uncertainty estimates than the currently widely used mean field factorisations. Part of this process involves an interesting recursive formulation of the problem which is likely to be of continued interest to future authors investigating structured DGP approximations. While the method proposed by the authors is very general, they successfully anchor its relevance and interest to the deep GP community by first highlighting the increased runtime demands of the fully-coupled model, and then providing a solution to this problem in the form of an explicit recommendation on how to construct a suitably structured covariance matrix. This is done by proposing a stripes-and-arrows sparsity pattern, which aims to capture the most important empirical patterns of the fully structured covariance matrix, namely information sharing between layers, as well as connecting each layer to the final output layer. By providing an explicit instance of a structure fully-coupled variational family, and by demonstrating the general feasibility of fully couple variational approximations, it is reasonable to expect that this paper can serve to inspire more research in this area to further move beyond the restrictions of mean-field factorisations and deliver improved uncertainty quantification in DGP models. The theoretical and methodological recommendations are supported by experimental evidence in Table 1 suggesting that this approach leads to performance that can deliver improved performance, certainly relative to the SGHMC method. Also important are the runtime experiments in Figure 3 which validates the theoretical computational complexity suggesting that this improved uncertainty quantification can indeed be achieved at an acceptable increase in runtime.

Weaknesses: The reported experiments are not entirely convincing that the extra work of this approximation has provided us with a substantially better model than that which was already available from the mean field factorisation. Only one experiment in Table 1 (protein) has a bold result for STAR DGP and not the MF DGP, and it is my understanding that performance is measured relative to the SGP, when perhaps it would have been more illuminating to measure performance relative to the MF DGP. Given the authors’ positive results on the relatively modest increase in computational complexity of the STAR DGP the seemingly modest increase in performance is not necessarily problematic, but it does suggest that perhaps more comment is needed on what exactly we gain from this form of structured covariance matrix. Is it possible that by linking the interior layer covariance to the output layer we have achieved something like a variance reduction in the interior layers, but that the diagonal structure of the interior layer covariances is still overly restrictive? Obviously these are not easy questions to answer, but some more intuition on this point would be useful, whether by further discussion or perhaps visualisation on simpler problems – for example one suggestion might be to fit a multi-layer DGP on a simpler simulate problem with several layers using the full covariance, the mean field and the STAR and then plot samples of the inner layers from the respective learned posteriors. This may help to better understand qualitatively what the STAR proposal gains over the MF and importantly what is perhaps still absent compared to the full covariance specification.

Correctness: Claims seem correct and well supported

Clarity: I found the paper well written and easy to follow

Relation to Prior Work: Yes -- the paper is well contextualised referencing both popular mean-field method, and recent work such as stochastic gradient HMC methods in this area.

Reproducibility: Yes

Additional Feedback:


Review 2

Summary and Contributions: The manuscript discusses the variational approximation over the inducing variables in deep Gaussian process models. Most often, this variational approximation is chosen to be a factorised multivariate Gaussian, which does not take into account the correlations between the layers of the deep GP and leads to poor predictive uncertainties. In the manuscript, a fully-coupled covariance for the variational approximation is chosen which enables capturing the correlation between layers and improves the predictive uncertainties. A general inference scheme is proposed to deal with this fully-coupled distribution in a computationally efficient manner. Finally, a specific structured covariance matrix that allows for correlations between subsequent layers is discussed in more detail.

Strengths: The inference scheme for the fully-parameterised Gaussian variational approximations over the global latent variables has two main benefits: 1) it is generally applicable and can thus easily be adapted to any other (domain-specific) structured covariance, 2) it allows to marginalise out the global variables (i.e. inducing points) -- lowering the stochasticity of the ELBO estimator. Both advantages are advancing the field of deep Gaussian processes and I believe will incentivise future work. Another strength of the manuscript is the constructive and detailed proof of Theorem 1. While the setup and idea is relatively simple (i.e. replace the mean-field variational approximation by a fully-coupled distribution), the derivations of the theorem are tedious and non-trivial. I think the deep GP community will benefit from having them written out so carefully and I have not seen them before.

Weaknesses: While in theory we would expect the structured covariances to improve the overall performance of deep GPs, I find the experimental evaluation not convincing. In particular: - Table S2: Apart from on the Boston dataset does the L2 SGHMC beat the L2/L3 STAR DGP. - Table S2: The STAR DGP is never better (confidence interval included) than the MF DGP. - Table 1: Apart from on the Naval dataset, is the STAR DGP never beating (confidence intervals taken into account) the MF DGP. I wonder if the authors have an explanation for this unexpected behaviour? I think a fairer description of the method’s performance in the text corresponding to the experiment is justified. I have some reservations regarding the experimental setup: - I have never seen the chosen train-test split based on a 1D projection used for Table 1 (extrapolation) before in the literature and -- while I understand the rationale behind it -- I find it quite arbitrary. This makes it hard to compare values across other work in this area. I believe that Table S2 should be the main table of the experiment and included in the main paper as I think there the standard conventions for UCI datasets are followed. See for example: https://github.com/hughsalimbeni/bayesian_benchmarks. - Given the limited amount of inducing points (128), layers (2 and 3) and number of outputs per hidden layer (5) I believe it should be possible to include a fully-coupled variational approximation to the baselines. This will give insight into the upper limit on the performance that can be achieved with the full model -- taking all correlations into account. While I appreciate that the STAR structure is just one configuration of the proposed method, I think its usefulness depends highly on the dataset. A more detailed analysis of different structures depending on the type of data would be a strong addition to the work and show the true power of the general framework developed in this paper.

Correctness: I am fairly confident that the claims in the paper are correct. My main concern with the paper in its current form is the empirical evaluation, as I have detailed above. I did not rigorously verify the author's proofs provided in the supplement.

Clarity: I enjoyed reading the paper: the writing is of a good standard, and the overall structure of the paper was easy to follow. If anything I would shorten the background section, and especially section 2.2 as this has been discussed in many previous works. This would leave more space to detail the proof of Theorem 1 (main contribution of the paper) as this is currently moved to the supplementary material. Furthermore, section 3.3 could also be discussed in the context of related work as it feels quite disjoint from other sections of section 3.

Relation to Prior Work: The problem this work addresses are put well into context with respect to other work in this area. The most important recent advances on inference in deep Gaussian processes are included in the related work sections of the manuscript.

Reproducibility: Yes

Additional Feedback: The paper claims that marginalising out the global variables leads to a more efficient model as it converges faster. While this is most probably correct, it is not proven nor empirically evaluated that this is indeed the case. Given that this is the main contribution of the paper, I would like to see a (toy) experiment where deep GPs with and without marginalisation of inducing variables are compared. Ideally, this could be a plot that shows the ELBO of both models as a function of time. A few minor additional comments/typos: - L186: double “we” - L188: “need”? - Capitalisation of reference titles is not consistent (e.g. gaussian, processes, inference, ...). ***** Post rebuttal ******* Thank you for the rebuttal and additional experiments. I increased my score accordingly.


Review 3

Summary and Contributions: The authors are concerned with variational inference in deep GPs and propose to models correlations in the posterior between inducing points of different layers. They demonstrate that the enriched posterior attains better predictive uncertainties at a slight increase in computational cost, positioning itself between the mean-field approximation and a full Gaussian approximation.

Strengths: The paper proves a recursion for the marginals of the fully-coupled variational approximation, which is novel. Moreover, they propose a new parameter-efficient parametrisation of the full covariance that still models some correlations between layers, which is also novel. The parametrisation is tested empirically and compared against other approaches. The authors carefully consider the computational aspects of their approach. They also implement a number of unit tests for the implementation (see e.g. lines 198-199), which is great and should be done more.

Weaknesses: I think this is a good paper, but there are two weaknesses: 1. On the UCI data set, the STAR approximation seems to not generally get significantly better results than the mean-field approximation. I would like to see a more thorough comparison, perhaps on other data, that teases apart exactly when the STAR approximation outperforms the mean-field approximation. 2. Although I am 90% conviced that the result of Theorem 1 is correct, the proof is hard to follow and I have not been able to fully convince myself that it is correct. To help the reader, I would like to see an interpretation of what the various quantities in Eqs (10) and (11) mean. For example, if I am not mistaken, \tilde{S}_n^{l l'} is the covariance between f_l and f_{l'} if the inputs in the DGP layers are fixed, but stochasticity due to the inducing points is retained (*). Moreover, it would be great if the sketch could motivate why the form of Gaussian conditionals in (10) and (11) appears (even though they are not really Gaussian conditionals, as is explained in the main text). Perhaps a heuristic argument along the following lines can be made: The conditional for f_n^L fixes the inputs of f_n^L and all previous DGP layers, because that is what conditioned on. Hence suppose that the inputs fixed, so not random, and consider the outputs of the layers, which are still random. The remaining stochasticity is due to the inducing points, and one obtains precisely the conditionals (10) and (11) by interpreting \tilde{S}_n^{l l'} as as in (*). The subtlety in this reasoning is that the outputs and inputs are considered separately, but in the model they are the same (outputs of previous layers are inputs to next layers). I wonder if the proof can be simplified with an argument along these lines. 3. The authors provided an implementation of the model, which is great, but did not provide code to reproduce the experiments. This makes it hard to double check the validity of experimental results.

Correctness: I believe that the claims are correct. Could the authors detail how the test log-likelihoods are computed? Are those joint or marginal probabilities?

Clarity: The paper is very well written.

Relation to Prior Work: Yes

Reproducibility: Yes

Additional Feedback: I appreciate the detailed appendix, which is helpful. EDIT: The additional results in the rebuttal are helpful. I maintain my original evaluation that this is a good submission.


Review 4

Summary and Contributions: Deep Gaussian processes improve the representational capability over GP through cascaded multiple GPs. It however should resort to the mean-field variational inference in order to achieve high computational efficiency. Hence, this paper presents a novel Gaussian variational family for retaining covariances among latent processes while achieving fast convergence by marginalizing out all global latent variables. The model has been deeply analyzed and verified on well-known benchmarks.

Strengths: The dependencies among the inducing variables over layers have been taken into account to improve the quality of variational posterior approximation.

Weaknesses: The empirical improvement of the proposed model over state-of-the-art DGPs is not significant. This may be caused by (i) the simplified covariances in/over layers to achieve reasonable computational efficiency; and (ii) the inconsistence between the coupled variational posterior and the factorized prior on inducing variables.

Correctness: The claims and method in this paper sound good.

Clarity: The paper is easy to follow and understand.

Relation to Prior Work: The difference has been clearly discussed.

Reproducibility: Yes

Additional Feedback: (1) As for the calibration case in Fig 2, the authors used the difference to the mean square error as a criterion to measure the quality of prediction variance. Is this reasonable? Is the mean square error a good base line to quantify the quality of variance? (2) The comparative results on the UCI benchmarks in Table S2 indicate that increasing the number of layers (L2 to L3) brings almost no improvement for the proposed model. The authors are suggested to explain more about this.

[Author Response · NeurIPS 2020]

We thank all reviewers for their careful reading and their detailed and constructive comments. We appreciate the positive feedback of all reviewers testifying our approach to be "advancing the field of deep Gaussian processes" (R2) and to inspire future research in this area (R1,R2). In particular, we want to thank the reviewers for acknowledging the constructive and detailed proof (R2), the easy to follow, well-written and well-contextualised manuscript (R1-R4), the careful consideration of computational aspects (R1,R3), and the helpful, detailed appendix (R3).

We first address the shared reviewer comments and then individual ones. The paper will be revised accordingly taking also further minor comments and suggestions of the reviewers into account.

*Empirical evaluation (R1-R4)* We agree with the reviewers that the presentation of the results was not entirely convincing. This is mainly due to the random 1D-projection of the extrapolation experiment: The direction of the projection has a large impact on the difficulty of the prediction task. Since this direction changes over the repetitions, the corresponding test log-likelihoods vary considerably, leading to large standard errors that hampered the comparison between the methods. We resolved this by performing a direct comparison between MF and STAR DGP as proposed by R1: To do so, we computed the frequency of test samples for which STAR DGP obtained a larger log-likelihood than MF DGP on

| Dataset | MF vs. STAR | MF vs. FC |
|---|---|---|
| boston | **0.55 ± 0.04** | **0.58 ± 0.04** |
| energy | **0.73 ± 0.05** | **0.70 ± 0.05** |
| concrete | **0.57 ± 0.04** | **0.56 ± 0.02** |
| wine red | **0.57 ± 0.04** | **0.57 ± 0.03** |
| kin8nm | *0.36 ± 0.03* | **0.59 ± 0.02** |
| power | 0.44 ± 0.06 | **0.68 ± 0.03** |
| naval | **0.67 ± 0.06** | *0.24 ± 0.07* |
| protein | 0.49 ± 0.03 | 0.50 ± 0.01 |

each train-test split independently. Average frequency $\mu$ and its standard error $\sigma$ were subsequently computed over 10 repetitions. On 5/8 datasets STAR DGP significantly outperforms MF DGP ($\mu > 0.50 + \sigma$), while the opposite only occurred on *kin8nm*. As suggested by R2, we also compared MF to FC DGP leading to similar results (see new table).

*Intuition for structured approximation (R1)* When we started working on the topic, we had the hypothesis that structured approximations would be especially helpful for test points that are distant from the training data and this idea also guided the layout of our experiments. While the results in our new table and Fig. 2 support our hypothesis, we were neither theoretically nor empirically able to pinpoint the underlying mechanism. We agree with R1 that an examination of inner layer samples for different structures (similarly as done in Ref. [34]) and the corresponding effects on the outputs are important research questions that need to be addressed in the future.

*Train-test split (R2)* We are the first to study the extrapolation behaviour of DGPs. While we agree that the splitting criterion could be improved, our experiments already reveal that established DGPs struggle in this setting. Furthermore, we indeed used the standard conventions for creating Tab. S2 and will move it to the main paper to facilitate comparison to related work.

*Convergence analysis (R2)* We thank the reviewer for proposing an empirical comparison of the convergence speed between analytical and MC marginalization. As proposed, we maximised the ELBO with both algorithms (using FC DGP L3 on the *concrete* dataset). We confirmed that the analytical marginalization converges quicker in terms of runtime (see new figure).

*Choice of structural approximation (R2)* In addition to the empirical motivation of our STAR structure (Fig. 1), the stripes pattern can also be justified from the model architecture: We expect the residual connections, realised by the mean functions (footnote 2), to lead to a coupling between successive latent GPs. In general, choosing the optimal structured approximation is highly model and data dependent. We agree that this is an important aspect of future work.

*Intuition for proof (R3)* We are amazed to find this heuristic argument in our reviews. While mathematically not rigorous, it provides the correct intuition. In fact, it was precisely the same reasoning that initially allowed us to come up with the induction hypothesis (Lem. 2). We will include this argument in the final version to provide additional guidance.

*Code and experiments (R3)* We thank the reviewer for the positive feedback on our unit tests. As suggested, we will also make the source code for the experiments publicly available. Test log-likelihoods were computed on the marginals.

*Inconsistency between coupled posterior and factorised prior (R4)* We agree that the role of coupled priors has not been thoroughly studied in deep GPs and should be investigated in more detail as it is done for the weight prior in Bayesian neural networks [e.g. Wenzel et al., ICML 2020].

*Relationship between variance and MSE (R4)* For a calibrated method, the predictive variance $\sigma_i^2$ is the expectation of the squared error ($\mathrm{SE}_i$) for test sample $i$. We estimated the latter by the empirical mean squared error (MSE) of test samples with a similar $\sigma_i^2$. The predictive variance $\sigma_i^2$ and the empirical $\mathrm{SE}_i$ are also compared in the test log-likelihood, $\log \mathcal{L} = -\frac{1}{2} \sum_i \left( \log \sigma_i^2 + \frac{\mathrm{SE}_i}{\sigma_i^2} \right)$, in which inaccurate predictions are penalised by the first and overconfident predictions by the second term (cf. the quantities in Fig. 2).

*Number of layers (R4)* We agree that the improvement of adding more layers (L2 to L3) in Tab. S2 is only significant for the *protein* dataset. However, this is in line with the results published in [24, Tab. 7], where the largest improvement is also observed on *protein*, and the only other dataset with significant but considerably smaller improvement is *kin8nm*.

[Meta-Review · NeurIPS 2020]

All the reviewers agreee that the paper proposed an interesting idea wrt structured variational approximations in DGPs but raised concerns with regards to the experimental evaluation. Most of these were addressed in the rebuttal and, in agreement with the reviewers, I recommend acceptance.